# Two orthogonal differentiation gradients locally coordinate fruit morphogenesis

Andrea Gómez-Felipe [1], Elvis Branchini [1], Binghan Wang [1], Marco Marconi [2,3], Hana Bertrand-Rakusová[1], Teodora Stan [1], Jérôme Burkiewicz [1], Stefan de Folter [4], Anne-Lise Routier-Kierzkowska [1], Krzysztof Wabnik [2,3] & Daniel Kierzkowski [1] ✉

Morphogenesis requires the coordination of cellular behaviors along developmental axes. In plants, gradients of growth and differentiation are typically established along a single longitudinal primordium axis to control global organ shape. Yet, it remains unclear how these gradients are locally adjusted to regulate the formation of complex organs that consist of diverse tissue types. Here we combine quantitative live imaging at cellular resolution with genetics, and chemical treatments to understand the formation of *Arabidopsis thaliana* female reproductive organ (gynoecium). We show that, contrary to other aerial organs, gynoecium shape is determined by two orthogonal, time-shifted differentiation gradients. An early mediolateral gradient controls valve morphogenesis while a late, longitudinal gradient regulates style differentiation. Local, tissue-dependent action of these gradients serves to fine-tune the common developmental program governing organ morphogenesis to ensure the specialized function of the gynoecium.

Morphogens provide positional information to control growth and patterning during the development of multicellular organisms[1–3]. They coordinate developmental gradients across organ axes, which is critical for establishing a proper body plan during early embryogenesis and for the development of planar organs both in plants and animals[4–6]. In aerial organs of plants such as leaves or sepals, gradients of growth and differentiation are typically established early during development along the longitudinal axis of the primordium to govern their shape[7–9]. These gradients are proposed to be controlled by global organizers or biomechanical forces that coordinate cellular behavior at the organ level[8–12]. Alternatively, it has been shown that organ forms can be generated by local organizers regulating an early establishment of developmental axes followed by tissue-dependent developmental behaviors[13]. This local regulation also underlies the establishment of

new growth axes introduced locally through gene and hormone activities to produce protrusions such as leaflets or serrations at the leaf margin[13–15].

The gynoecium, the female reproductive structure of the flower, is composed in *Arabidopsis thaliana* of two carpels fused with the repla and topped with the style and stigma (Fig. 1c)[16,17]. Gynoecium patterning is established early during development and is tightly controlled by transcription factors and signaling molecules[18]. Phytohormones such as auxin and cytokinin, have been shown to play a crucial role in gynoecium patterning[19]. For instance, the disruption of auxin biosynthesis, transport, or signaling significantly affects the proper establishment of tissue types along both mediolateral and longitudinal axes of the gynoecium[20–26]. Despite this knowledge, it is unclear how the dynamics of cellular growth and differentiation are

[1]Institut de Recherche en Biologie Végétale, Département de Sciences Biologiques, Université de Montréal, 4101 Sherbrooke St E, Montréal, QC H1X 2B2, Canada. [2]centro De Biotecnología Y Genómica De Plantas (Universidad Politécnica De Madrid (Upm), Instituto Nacional De Investigación Y Tecnología Agraria Y Alimentaria (Inia, Csic), Campus De Montegancedo, Pozuelo De Alarcón, 28223 Madrid, Spain. [3]Departamento de Biotecnología-Biología Vegetal, Escuela Técnica Superior de Ingeniería Agronómica, Alimentaria y de Biosistemas, Universidad Politécnica de Madrid (UPM), Madrid 28040, Spain. [4]Unidad de Genómica Avanzada (UGA-LANGEBIO), Centro de Investigación y de Estudios Avanzados del Instituto Politécnico Nacional (CINVESTAV-IPN), CP, 36824 Irapuato, Mexico. ✉e-mail: daniel.kierzkowski@umontreal.ca

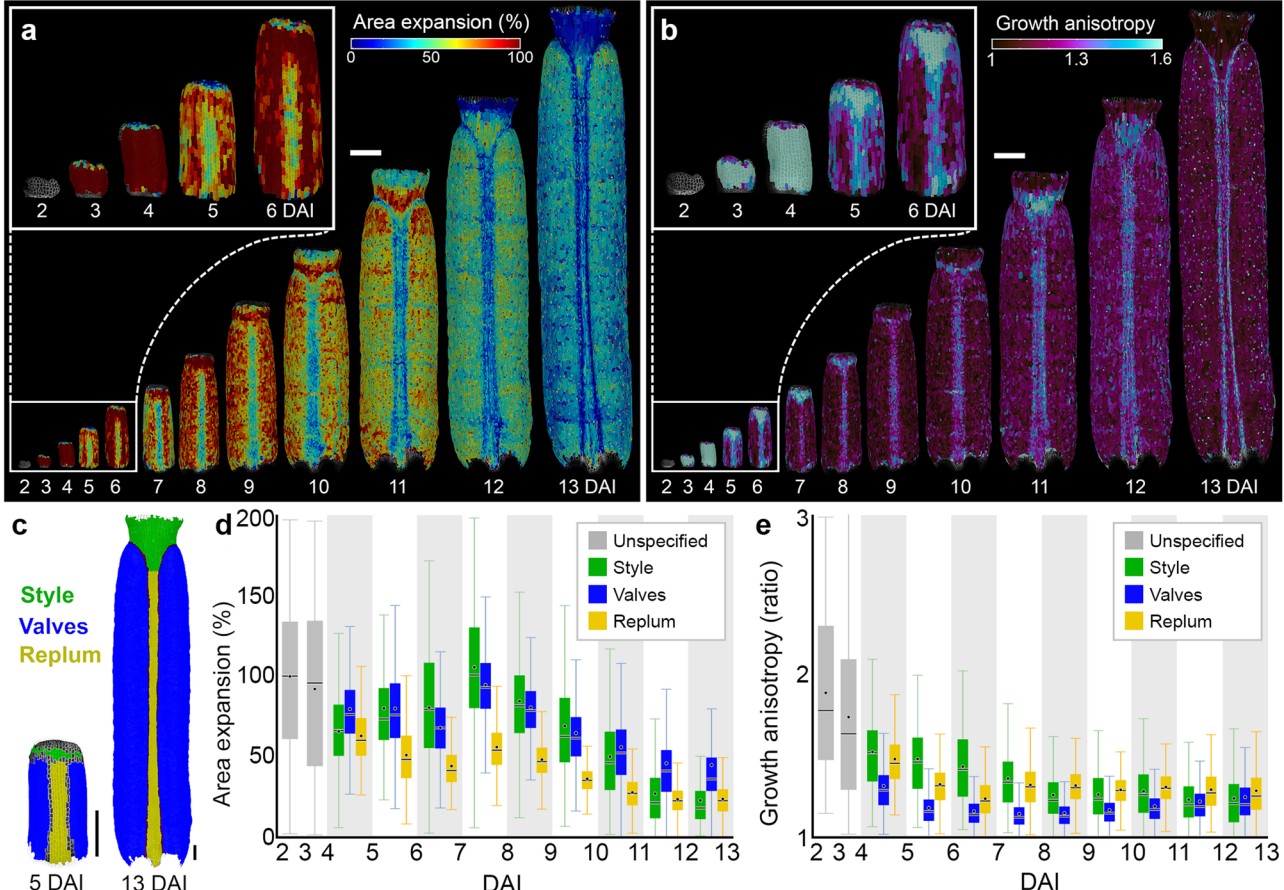

**Fig. 1 | Cellular growth patterns underlying gynoecium development. a, b,** Heatmaps of area expansion (**a**) and growth anisotropy (**b**) for the *Arabidopsis thaliana* gynoecium. **c** Lineage tracing of style (green), valves (blue), and replum (yellow) between 5 and 13 DAI. **d, e** Quantifications of area expansion (**d**) and growth anisotropy (**e**) in different regions of the developing gynoecium (*n* = 136 cells at 3 DAI; *n* = 346 cells at 4 DAI; *n* = 94 (style), 516 (valves), 201 (replum) cells at 5 DAI; *n* = 109 (style), 836 (valves), 347 (replum) cells at 6 DAI; *n* = 190 (style), 1421 (valves), 568 (replum) cells at 7 DAI; *n* = 298 (style), 2459 (valves), 913 (replum) cells at 8 DAI;

*n* = 432 (style), 3921 (valves), 1425 (replum) cells at 9 DAI; *n* = 584 (style), 7039 (valves), 2078 (replum) cells at 10 DAI; *n* = 727 (style), 10,933 (valves), 2789 (replum) cells at 11 DAI; *n* = 767 (style), 14,579 (valves), 3545 (replum) cells at 12 DAI; *n* = 532 (style), 10,779 (valves), 2778 (replum) cells at 13 DAI; three independent time-lapse series). The boxplots represent a range between the first and the third quartile and the whiskers include 95% of the values. Lines represent the median and dots represent the mean. DAI days after gynoecium initiation. Scale bars, 100 μm. See also Supplementary Fig. 1 and Supplementary Movies 1 and 2.

regulated in different regions of the gynoecium and how they are influenced by tissue identities.

The current model suggests that the growth of the gynoecium is governed globally by a polarity field spanning the epidermis and oriented parallel to the organ's longitudinal axis[27]. However, in contrast to other aerial organs such as leaves or sepals, gradients of growth and cell differentiation have not been reported in the *Arabidopsis thaliana* gynoecium[27,28]. This remarkable exception is surprising as carpels have been shown to have a leaf-like origin[29]. It raises the question of how a common developmental program, likely governing organ shaping in plants[13,30–32], is modified to control gynoecium morphogenesis. Local regulation of cellular growth and differentiation could overwrite typical organ-wide developmental gradients and ensure the specialized function of this complex organ in plant reproduction.

## Results

### Cellular growth patterns underlying gynoecium development

Gynoecium develops inside the enclosed floral bud, and it is very challenging to image and measure its growth, thus hindering our understanding of underlying developmental processes. To overcome this limitation, we developed a new imaging method to precisely follow the cellular dynamics guiding gynoecium formation in *A. thaliana* (see Methods). We recorded the cellular behavior from the early stages

at 2 days after gynoecium initiation (2 DAI, equivalent of floral stage 7) until its final shape was established at 13 DAI[33]. Using MorphoGraphX software[34], we extracted growth dynamics for all cells in the surface layer of one side of the gynoecium (from ~70 cells at early primordium to ~11,000 cells at anthesis), and quantified growth rates, growth anisotropy (the ratio of expansion in the maximal and minimal principal directions of growth), cell divisions, and cell geometries (i.e., cell sizes) (Fig. 1a, b, Supplementary Fig. 1a, b, Supplementary Movies 1 and 2). We used lineage tracing[13] to determine the origin of the main gynoecium regions (i.e., valves, style, replum) and quantified their cellular growth parameters (Fig. 1c–e, Supplementary Fig. 1c, d).

In the early stages (from 2 to 4 DAI), cellular growth and proliferation were high, and the gynoecium expanded along its longitudinal axis (Fig. 1a, b,d-e, Supplementary Fig. 1a, c, Supplementary Movies 1 and 2). Between 4 and 5 DAI, regions that will develop into valves, replum, and style started to display divergent growth behaviors. The future style continued to elongate rapidly, while cell growth in the replum strongly decreased but maintained its longitudinal orientation. Valves continued to grow relatively fast but temporarily transitioned to more isotropic growth (Fig. 1). Consistent with previous studies[27,28], cell growth within both replum and valves appeared to be relatively uniform. By contrast, we observed a non-homogeneous growth in the style (from 9 DAI), with cells located at the tip growing slower than at its base (Fig. 1a). This growth gradient in the style seems

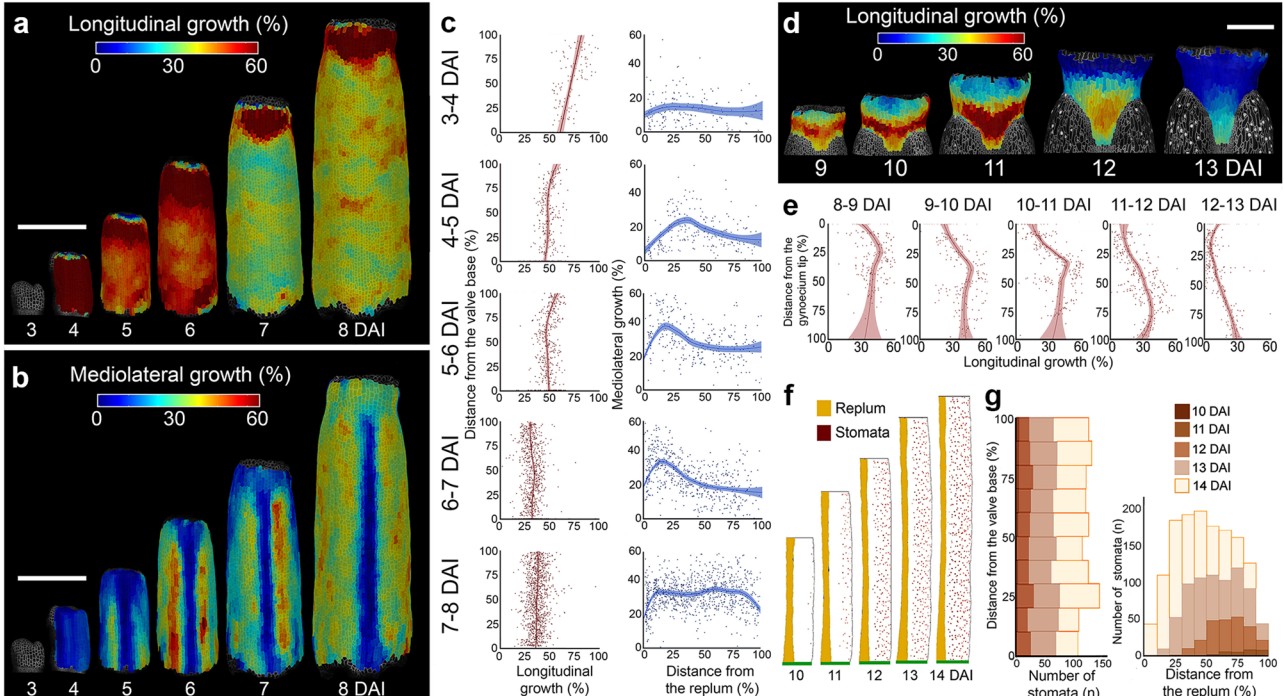

**Fig. 2 | Two orthogonal gradients occur during gynoecium development.**
**a**, **b** Heat-maps of the averaged cellular growth along longitudinal (**a**) and medio-lateral (**b**) axes of the gynoecium. **c** Quantification of the cellular growth along the longitudinal (left) and mediolateral (right) axis of the valve as a function of the distance from the valve base or from the replum ($n = 137$ cells at 3–4 DAI; $n = 283$ cells at 4–5 DAI; $n = 408$ cells at 5–6 DAI; $n = 789$ cells at 6–7 DAI; $n = 1365$ at 7–8 DAI). **d** Heat-maps of averaged cellular growth along the longitudinal axis of the style. **e** Quantification of the cellular growth along the longitudinal axis of the style as a function of the distance from the gynoecium tip ($n = 207$ cells at 8–9 DAI; $n = 207$ cells at 9–10 DAI; $n = 258$ cells at 10–11 DAI; $n = 297$ cells at 11–12 DAI; $n = 298$ cells at 12–13 DAI). **f** Stomata distribution in the valves. Replum in orange, valve in white, stomata in brown, base of the valve in green. **g** Quantification of stomatal distribution as a function of the distance from the valve base (left) or from the replum (right) ($n = 6$ stomata at 10 DAI; $n = 58$ stomata at 11 DAI; $n = 517$ stomata at 12 DAI; $n = 605$ stomata at 13 DAI; $n = 1209$ stomata at 14 DAI; three independent time-lapse series). DAI days after gynoecium initiation. For scatter plots, the distance was normalized. Dots in **c**, **e** indicate each growth value, lines represent the average and shaded areas represent standard deviation (SD). Scale bars, 100 µm in (**a**, **b**) and 50 µm in (**d**). See also Supplementary Figs. 1 and 2 and Supplementary Movie 3.

to briefly invade the very top of the valves where we observed a transient increase of growth rates at 11 DAI (Fig. 1a).

These findings indicate that the common basipetal (from the tip to the base) developmental gradient observed in all *Arabidopsis* aerial organs is also present in the gynoecium, but it is significantly delayed during development and is mainly restricted to the style. This restriction suggests that cellular growth and differentiation may be locally and independently coordinated across different parts of the developing gynoecium at different developmental windows.

### Two orthogonal gradients occur during gynoecium development

To verify this scenario, we next computed cellular growth rates in different regions of the gynoecium along its longitudinal and medio-lateral axes (Fig. 2a, b, Supplementary Fig. 2). The longitudinal growth was largely homogeneous in the valves except for a transient increase at their tips at 11 DAI (Supplementary Fig. 2a). By contrast, we observed a clear longitudinal gradient of growth in the style from 9 DAI (Fig. 2d, e). This gradient of growth was accompanied by the progression of cell differentiation (measured as an increase in cell sizes) from the tip to the base of the style (Supplementary Fig. 1b, e). Surprisingly, we recorded unequal cellular growth along the mediolateral axis of the valves from 4–5 DAI (Fig. 2b, c). Specifically, valve cells located close to the replum increased their growth, establishing a clear gradient of cellular expansion along the mediolateral axis (Fig. 2b, c). This gradient persisted until 7–8 DAI when growth along the mediolateral axis largely equalized in the valves (Fig. 2b, c, Supplementary Fig. 2b, Supplementary Movie 3). Importantly, we also observed a

corresponding gradient of cellular differentiation revealed by the patterning and appearance of stomata and an increase of cell sizes occurring first in the more lateral regions of the valves (Fig. 2f, g, Supplementary Fig. 1b, f–h). Our data suggest that two orthogonal gradients of growth and differentiation control gynoecium morphogenesis along its longitudinal and mediolateral axes. These gradients appear to be tissue-dependent, as they operate locally along different axes in distinct regions of the organ. Importantly, they are time-shifted, with cell differentiation in the valves starting already at around 6 DAI while in the style only around 9 DAI.

### Auxin patterning during gynoecium development

The molecular cues that could set these gradients are largely unknown. In theory, a longitudinal gradient may emerge from a hypothetical signal diffusing from the organ base that would prevent cellular differentiation in more proximal regions of the organ[10,13]. However, this scenario is unlikely as the establishment of gradients in style occurs at late developmental stages (from 9 DAI) when the style is located far away from the base of the organ. Alternatively, the longitudinal gradient could be controlled by a small molecule originating from the organ tip that would stimulate cell differentiation. The phytohormone auxin is a plausible candidate for such a signal, as auxin is known to control both cell growth and differentiation through concentration gradients in plants[13,35–37]. Moreover, auxin was also suggested to control gynoecium patterning along its main axis[25].

To test this hypothesis, we first monitored auxin responsiveness in the gynoecium using the *pDR5v2::nls-3xVenus* reporter line[38]. *DR5v2* signal in the epidermis was localized at the very tip of the gynoecium

until anthesis, with its intensity decreasing progressively at late developmental stages from around 7–8 DAI (Fig. 3a, Supplementary Fig. 3a). Auxin synthesis monitored with *YUCCA4* (*pYUC4::3xNLS-GFP*) reporter line[39] was initially detected in all epidermal cells (at 2 DAI), then became restricted to the replum (3–5 DAI) and the very tip of the style (3–10 DAI) and was finally eliminated from the epidermis at around 11 DAI (Fig. 3c, Supplementary Fig. 3b). The PIN-FORMED1 (PIN1) auxin efflux carrier was present at the upper membranes in all epidermal cells of the gynoecium at early stages (2–3 DAI) and later was progressively restricted to the replum and the style, consistent with its involvement in the apical accumulation of auxin[40] (Fig. 3e–g). Interestingly, the first signs of the establishment of the longitudinal gradient of growth in the style (from around 9 DAI) coincided with the elimination of the PIN1 expression from the epidermis (Fig. 1a, Fig. 2d, Fig. 3e, Supplementary Fig. 3c). This suggests that acropetal (from the base to the tip) auxin transport in the style could help restrict auxin to its tip at early developmental stages to prevent precocious epidermal cell differentiation. At later stages, when PIN1 is absent from the epidermis, auxin could move through other PINs to more proximal regions triggering the basipetal gradient of cell growth and differentiation.

To explore this possibility, we compared the expression of other PIN proteins involved in gynoecium patterning with growth and cell differentiation gradients in the style. PIN3 was initially (until 8 DAI) expressed at the tip of the style in a non-polarized manner consistent with its role in style radialization (Fig. 3h)[24,40]. Strikingly, from 9 DAI, the PIN3 expression domain started to extend toward more proximal regions of the style, tightly following the gradient of cellular growth in this region (Fig. 3h, i). Importantly, PIN3 appeared to switch its polar localization away from the organ tip (Fig. 3i), suggesting the progressive establishment of basipetal auxin flow through the style at late developmental stages. Consistently, PIN7 followed a comparable pattern of distribution and polarization in the style epidermis from 9 DAI (Fig. 3j, k). These data indicate that auxin originating from the gynoecium tip may set up the longitudinal gradient of growth and differentiation in the style at late developmental stages.

How could an early mediolateral gradient be established in the valves after their emergence? During early gynoecium development, a carpel margin meristem (CMM) is formed along the margins at the inner side of the fused carpels. The CMM gives rise to the placenta which progressively produces ovules[41–43] and is critical for the establishment of the valves[19,24]. The activity of CMM monitored by the ovule primordia initiation and associated auxin production (*YUC4* expression), auxin response (*DR5v2*), and PIN1-mediated auxin transport, coincided with the establishment of the mediolateral gradient of growth and differentiation in the valves from around 4 to 7 DAI (Fig. 2b, Fig. 3b, d, f, Supplementary Fig. 4)[43]. Therefore, we hypothesize that an active CMM may be a source of a putative mobile signal that sets a mediolateral gradient of growth and differentiation in the valves by locally increasing growth rates and delaying an overall cell differentiation adjacent to the replum. A valve-specific polar auxin transport could also be involved in enhancing the mediolateral gradients as we have observed a clear mediolateral asymmetry of PIN7 expression in the valves starting from 9 DAI (Fig. 3j). Additionally, YUCCA2 gene was previously reported to be expressed in lateral parts of the valves[21]. Regardless of the precise mechanism, the differentiation gradients in the valves are set early in the development (at around 6 DAI), arising in the context of undifferentiated primordium well before the initiation of the longitudinal differentiation gradients that are characteristic of all lateral organs.

### Removing valves abolishes the mediolateral and expands the longitudinal gradients

If the mediolateral gradient of differentiation is valve-specific, the absence of the valves should result in the disappearance of this gradient. It has been shown that the polar auxin transport is involved in the mediolateral patterning of the gynoecium via the establishment of the procambium in the future carpels at early developmental stages[24]. We, therefore, abolished this valve patterning using transient naphthylphthalamic acid (NPA) treatment before organ initiation (one week before observations) and followed organ development for 10 days on the medium without NPA. This treatment led to the complete elimination of the valves and conversion of the gynoecium into a radially symmetric tube-like structure (Fig. 4a, b, Supplementary Fig. 5, Supplementary Movie 4)[19,25]. In this valveless gynoecium, the mediolateral growth was abolished while the organ expanded longitudinally (Supplementary Fig. 5). As in the wild-type, we observed an establishment of the longitudinal gradient of growth and differentiation at later developmental stages (from 7 DAI). These gradients, however, extended toward more proximal regions of the organ as compared to the untreated wild-type where they were mainly restricted to the style (Fig. 4a–f, Supplementary Fig. 5a). Broadening of the longitudinal gradient is consistent, at least partially, with the previously suggested basal expansion of the style identity upon NPA treatment[25].

The onset of the longitudinal gradients happened after the complete elimination of the apically polarized PIN1 auxin efflux carrier from the epidermis (Fig. 4g–i, Supplementary Fig. 6a). Strikingly, PIN3 that was initially (until 5 DAI) localized at the tip of the gynoecium in a non-polarized manner, started (from 5–6 DAI) to polarize basally and progressively expanded throughout the entire epidermis at 9 DAI (Fig. 4j, k) that correlated with a basipetal progression of cell differentiation throughout the organ. This suggests that auxin, initially synthesized and accumulated at the gynoecium tip (Supplementary Fig. 6b–d), was likely transported through the epidermis toward the organ base to trigger the basipetal cell differentiation along the longitudinal axis of the valveless gynoecium. Thus, the lack of valve identity disables the ability to establish the mediolateral gradients in the gynoecium. This could, in turn, allow the establishment of the organwide basipetal gradient of cell differentiation in the valveless gynoecium.

### Introducing carpel identity into sepal reorients organ differentiation gradients

The valve identity and the presence of the CMM seem to be important for the establishment of the early mediolateral developmental gradients. If this is the case, converting other types of lateral organs into a carpel-like structure with ectopic meristematic activity along their margins should switch their typical basipetal gradients operating along the longitudinal axes[9,10,13,44] toward mediolateral orientation as observed in the gynoecium (Fig. 2).

In wild-type sepals, a clear longitudinal gradient of cellular growth and differentiation was visible from around 4 DAI (Fig. 5a, c, d, g, Supplementary Fig. 7a–c, Supplementary Movie 5)[9]. We then analyzed growth in the *ap2-7* mutant sepals, which develop carpeloid structures including placenta and ovules along their margins while maintaining some typical sepal features such as giant cells, and planar organ geometry (Fig. 5b, e, f)[45]. In such carpelized sepals, the typical basipetal gradient of cellular growth and differentiation was largely eliminated (Fig. 5e, f, Supplementary Fig. 7d–f, Supplementary Movie 5). Furthermore, cells located close to the modified margin of the *ap2-7* sepal tended to differentiate later than cells further away from the margin, indicating that the meristematic activity along the carpelized sepals in this mutant led to the establishment of the mediolateral gradient of differentiation (Fig. 5f, h, Supplementary Fig. 7f). These findings support the idea that mediolateral gradients are tissue identity dependent and may be locally regulated by the activity of the CMM.

### Removing style identity and reducing CMM expands the longitudinal gradient

To further investigate the relationship between gradient dynamics and tissue identity establishment, we analyzed the development of the *crc-*

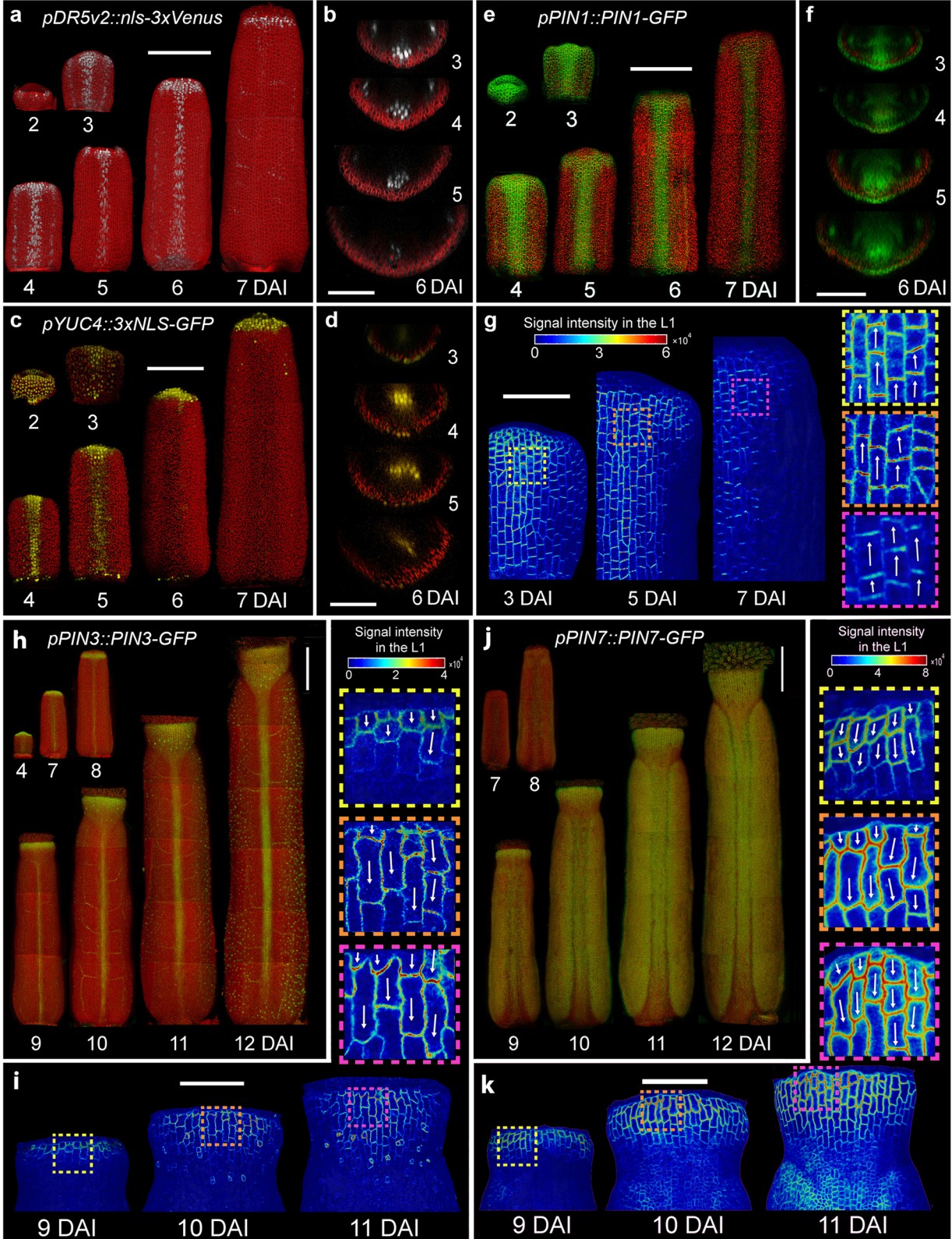

**Fig. 3 | Auxin patterning during gynoecium development. a–k** Expression patterns of *pDR5v2::nls-3xVenus* (**a**, **b**), *pYUC4::3xNLS-GFP* (**c**, **d**), *pPIN1::PIN1-GFP* (**e–g**), *pPIN3::PIN3-GFP* (**h**, **i**), and *pPIN7::PIN7-GFP* (**j**, **k**) in the *A. thaliana* gynoecium. Virtual cross sections shown in (**b**, **d**, **f**) are in the middle of the gynoecium. **g**, **i**, **k** Heat-maps represent the intensity of the *PIN1-GFP* (**g**), *PIN3-GFP* (**i**), and *PIN7-* *GFP* (**k**) signal in the L1 epidermal layer (one side of the gynoecium tops are shown for PIN1 and the style is shown for PIN3 and PIN7). Insets indicate membrane localization and arrows indicate plausible orientation of PIN polarization. Scale bars, 100 µm in (**a**, **c**, **e**, **h**, **j**) and 50 µm in (**b**, **d**, **f**, **g**, **i**, **k**). See also Supplementary Fig. 3.

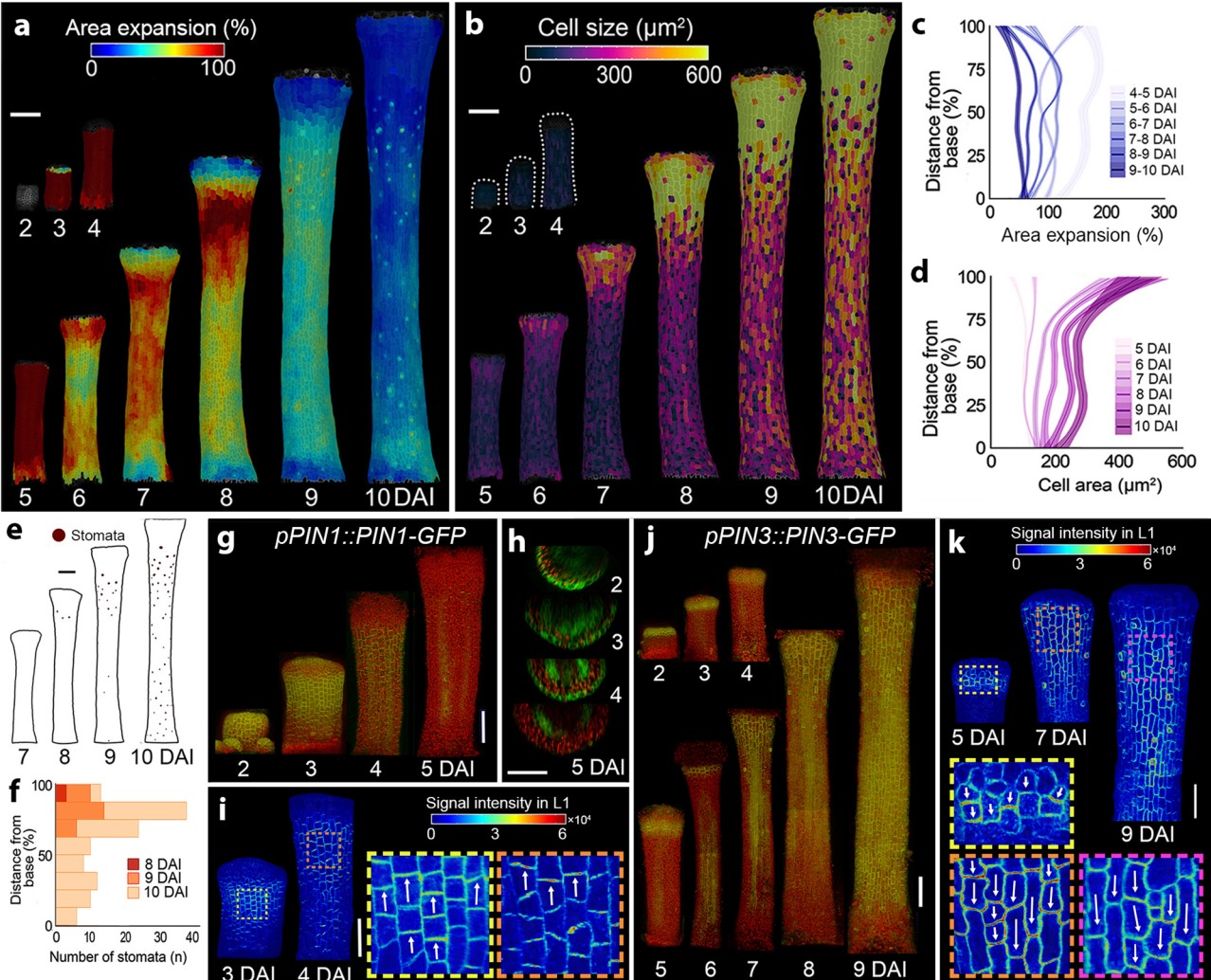

**Fig. 4 | Removing valves abolishes the mediolateral and expands the longitudinal gradients. a, b** Heat maps of averaged area expansion (**a**) and cell sizes (**b**) in *A. thaliana* gynoecium one week after treatment with the NPA. **c, d** Quantification of cellular growth (**c**) and cell sizes (**d**) as a function of the distance from the gynoecium base after NPA treatment (*n* = 889 cells at 5 DAI; *n* = 1234 cells at 6 DAI; *n* = 1460 cells at 7 DAI; *n* = 1463 cells at 8 DAI; *n* = 1476 cells at 9 DAI; *n* = 1478 cells at 10 DAI; three independent time-lapse series). **e, f** Stomata distribution (**e**) and quantification of stomatal distribution along the distance from the gynoecium base (**f**) after NPA treatment (*n* = 3 stomata at 8 DAI; *n* = 32 stomata at 9 DAI; *n* = 125 stomata at 10 DAI; three independent time-lapse series). **g–k** Expression patterns of *PIN1-GFP* (**g–i**) and *PIN3-GFP* (**j–k**) in gynoecia after NPA treatment. Heat maps represent the intensity of the *PIN1-GFP* (**i**) and *PIN3-GFP* (**k**) signals in the L1 layer. Insets indicate membrane localization; arrows indicate plausible orientation of PIN polarization. For plots, the distance was normalized, lines represent the average and shaded areas represent standard deviation (SD). DAI days after organ initiation. Scale bars, 100 μm (**a, b, g, h, j**) and 50 μm (**i, k**). See also Supplementary Figs. 5 and 6, Supplementary Movie 4.

*1 spt-12* double mutant (Fig. 6a–d). This mutant displays a split phenotype, with partially fused carpels lacking style identity and papilla, as well as decreased activity of the CMM, reflected by lower ovule production and a strong reduction of other medial-derived tissues (Fig. 6a)[46]. The distal, unfused portion of the organ displayed leaf-like characteristics including a pointy tip, formation of marginal serrations, and the absence of ovules along its margins (Fig. 6a). Interestingly, this part quickly established (from 5 DAI) a longitudinal gradient of growth and differentiation, spanning around half of the organ length at 8 DAI, (Fig. 6b, c, d, Supplementary Movie 6). Thus, removing style identity seems to largely accelerate the progression of the basipetal differentiation gradient in the distal part of the organ which now follows comparable timing and dynamics as in the leaf or sepal (Fig. 5c)[9,44].

In contrast, the bottom portion of the organ in the mutant resembled much more a wild-type gynoecium with two carpels partially fused by replum and producing ovules (Fig. 6a). In this basal region, we observed a clear mediolateral gradient of cell differentiation in the valves with the first stomata appearing away from the

reduced replum at 8 DAI (Fig. 6c, d). Thus, genetic restriction of the carpel identity and CMM domain to proximal regions in the double mutant, combined with the early onset of the basipetal gradient of cell differentiation in the absence of the style identity, allowed the expansion of the longitudinal gradient toward more proximal regions of the organ.

Taken together, our data indicates that gynoecium morphogenesis is controlled by two complementary orthogonal gradients of differentiation that act locally and are dependent on the local tissue identity. It also suggests that the timing of the onset of each gradient may influence the overall shape of the organ (Fig. 6f).

## Discussion
Our study has unveiled an underlying mechanism that shapes gynoecium form, modified from the common developmental processes observed in other aerial organs in plants. This mechanism is driven by the action of two orthogonal time-shifted differentiation gradients. The orientation and timing of these gradients do not seem to be

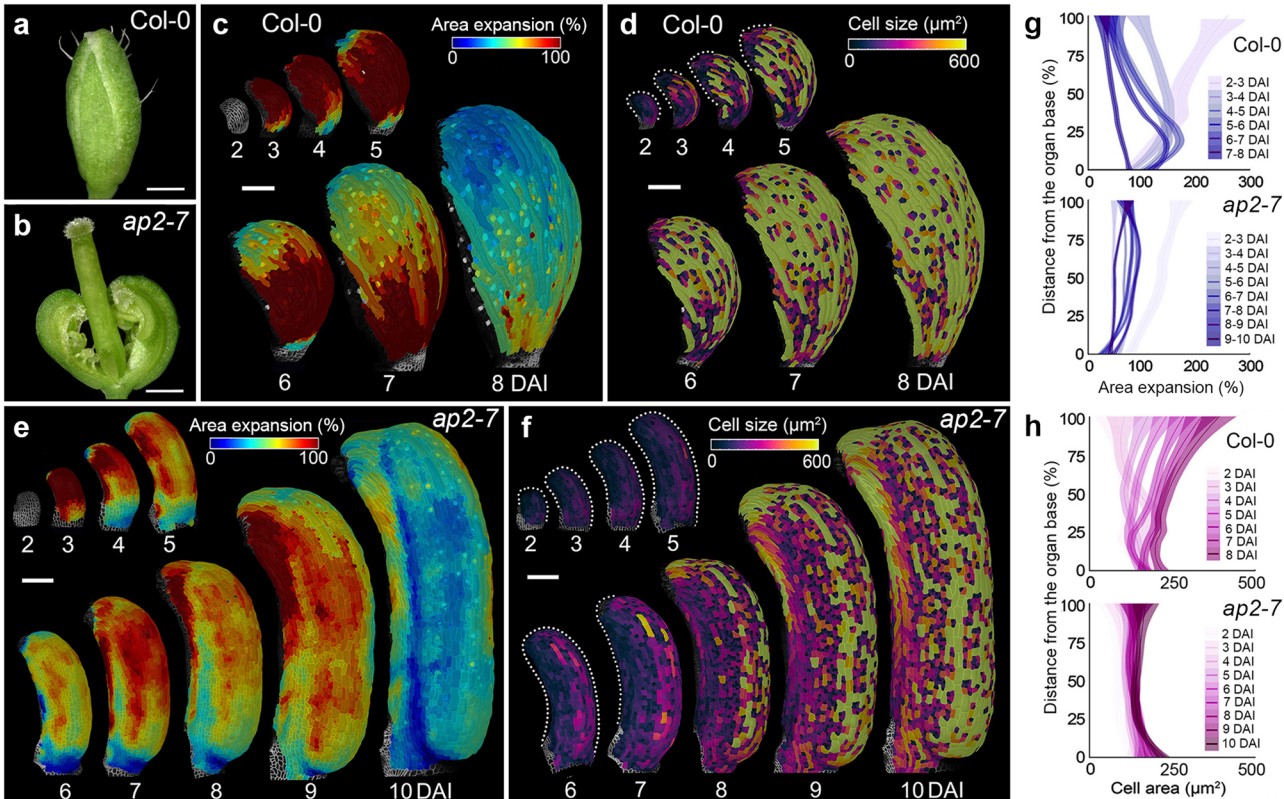

**Fig. 5 | Introducing carpel identity into sepal reorients organ differentiation gradients. a, b,** Wild-type (**a**) and *ap2-7* mutant (**b**) sepals. **c, f** Heat maps of averaged area expansion (**c, e**) and cell sizes (**d, f**) of wild-type (**c, d**) and *ap2−7* mutant (**e, f**) sepals. **g, h** Quantifications of area expansion (**g**) and cell sizes (**h**) of wild-type sepal (*n* = 354 cells at 3 DAI; *n* = 707 cells at 4 DAI; *n* = 832 cells at 5 DAI; *n* = 1803 cells at 6 DAI; *n* = 2457 cells at 7 DAI; *n* = 3145 cells at 8 DAI; three independent time-lapse series) and *ap2−7* mutant sepal (*n* = 73 cell at 2 DAI; *n* = 132 cells at 3 DAI; *n* = 375 cells at 4 DAI; *n* = 566 cells at 5 DAI; *n* = 881 cells at 6 DAI; *n* = 1299 cells at 7 DAI; *n* = 2135 cells at 8 DAI; *n* = 3279 cells at 9 DAI; *n* = 1294 cells at 10 DAI; three independent time-lapse series) along the longitudinal axis of the organ. For plots, the distance was normalized, lines represent the average and shaded areas represent standard deviation (SD). DAI days after organ initiation. Scale bars, 500 μm (**a, b**), 100 μm (**c–f**). See also Supplementary Fig. 7 and Supplementary Movie 5.

globally controlled at the organ level but rather depend on the local tissue identities. The mediolateral gradients are established in the valves at the early stages of the development while the longitudinal gradient is delayed and initiates much later in the style. A general delay in basipetal cell differentiation within the gynoecium may be a critical event for the proper patterning of the fruit. This delay could allow the introduction of the mediolateral gradient in the valves before the onset of the basipetal gradient. From an evolutionary perspective, the local reorientation of cellular differentiation in the valves may have contributed to the success of flowering plants by enabling the simultaneous maturation of ovules, a crucial factor for successful plant reproduction.

An intriguing concept emerging from this study is the potential restriction of the longitudinal gradient's progression to the style due to a relatively early onset of cell differentiation in the valves. In such a scenario, cells in the valves may have a limited ability to respond to the tip-derived signal, given their advanced state of differentiation along the mediolateral organ axes when the basipetal signal appears. Notably, the constitutive expression of NGATHA 3 and/or STYLISH 1 (genes involved in style identity) induces partial conversion of the ovary into style tissue. This leads to increased growth of the valve's tips (more pronounced shoulders), perturbed longitudinal patterning of the gynoecium, and reduced plant fertility[47]. Whether such gradient interactions regulate gynoecium development requires further investigations.

Also, we predict two plausible signals that could control orthogonal gradients of differentiation, one likely being auxin itself[24,25,40,48]. Our data suggests that auxin initially restricted to the tip of the

gynoecium, could be a main signal triggering basipetal gradient of differentiation throughout the style. Interestingly, a transient acceleration of growth at the valve shoulders (Supplementary Fig. 2a) may suggest that auxin transported through the style could reach the top part of the valves locally triggering growth (Supplementary Fig. 2a). Modulation of such mechanism could play a role in valve shape transformations in other species from Brassicaceae family, such as *Capsella rubella*. In this species auxin is known to accumulate at the tip of the valves that form pronounced shoulders post fertilization[49].

The second signal could be another type of small molecule that is known to act in a concentration-dependent manner including plant hormones[50–52]. Cytokinin emerges as a particularly plausible candidate for such a signal, given its activity in the CMM, its ability to diffuse through the tissue, and its known role in delaying cell differentiation[19,26]. Upon NPA treatment, cytokinin signaling increases all over the central region of the radialized gynoecium which may be the reason why cells don't differentiate until they receive auxin derived from the tip (Fig. 4)[19]. Auxin could also be implicated in the establishment of the mediolateral gradient of differentiation as it has been shown to be synthesized in the lateral parts of the valves and polar auxin transport underlie the early patterning of vascular tissue that is critical for valve formation[21,24]. Finally, the role of mediolateral signaling is also supported by a recent study showing that a putative signal derived from the developing seeds is critical for fruit expansion after fertilization[28].

There are interesting similarities between the proposed mechanism with morphogen gradients operating along a single axis of growth in synthetic and natural systems[53,54]. However, unlike other studies, this

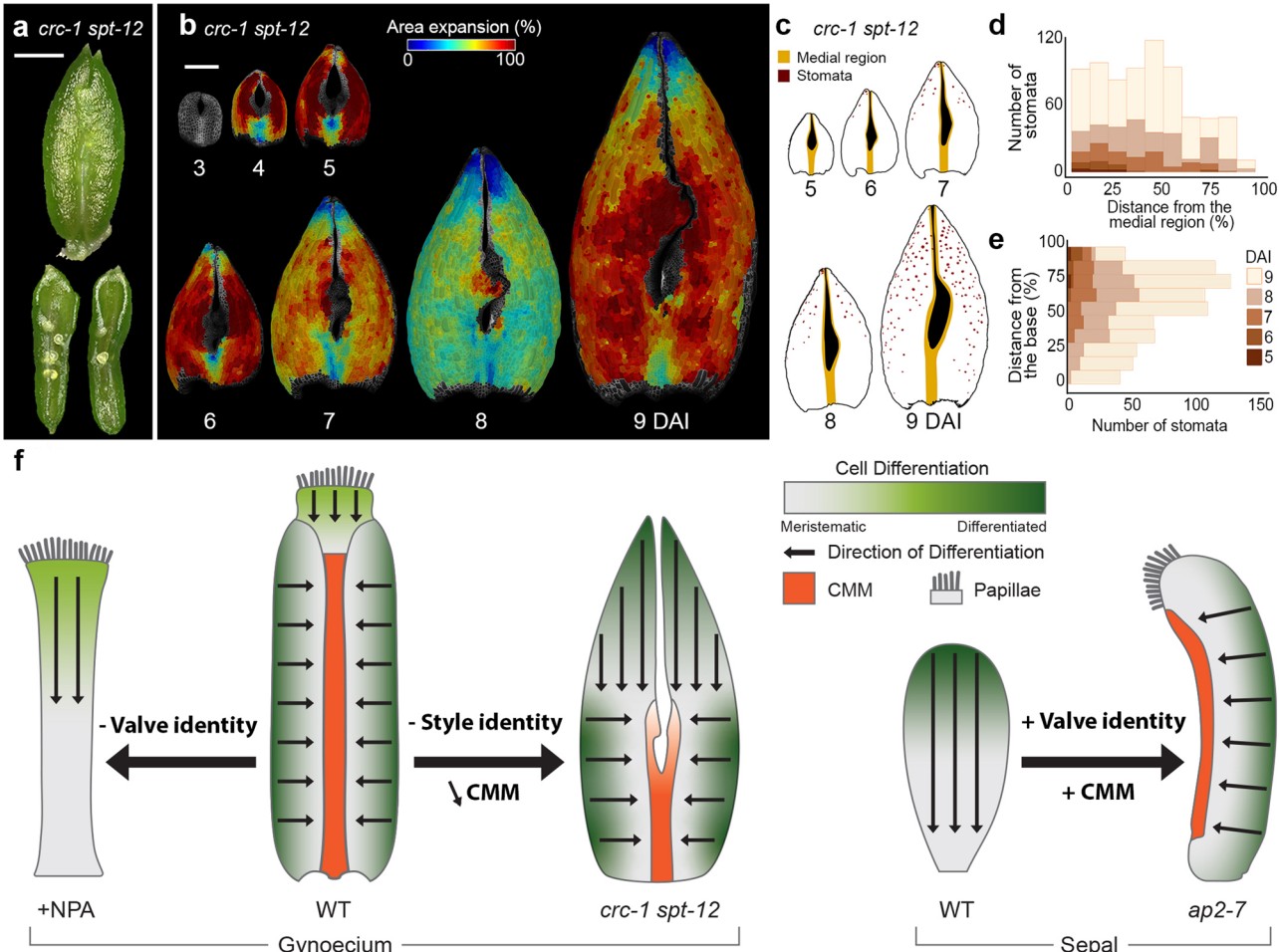

**Fig. 6 | Removing style identity and reducing CMM expands the longitudinal gradient. a** Gynoecium of the *crc-1 spt-12* double mutant at 10 DAI (top); longitudinal cut (bottom) shows ovule-like structures. **b** Heat maps of averaged area expansion of *crc-1 spt-12*. **c** Stomata distribution of *crc-1 spt-12*. **d, e** Quantification of stomatal distribution as a function of the distance from the valve base (**e**) or from the medial region (**d**) of *crc-1 spt-12* (n = 6 stomata at 5 DAI; n = 15 stomata at 6 DAI; n = 61 stomata at 7 DAI; n = 309 stomata at 8 DAI; n = 815 stomata at 9 DAI; three independent time-lapse series). **f** Schematic representation of the influence of the tissue identity and marginal meristematic activity on developmental gradients. Removing valve's identity (after NPA treatment) abolishes the mediolateral and expands the longitudinal gradients. Removing style identity and reducing CMM (in *crc-1 spt-12* mutant), accelerates and expands the longitudinal differentiation gradient. Introducing valve identity and CMM into sepal reorients organ differentiation gradients. Orange indicates CMM; green, cell differentiation status; arrow, the orientation of cell differentiation progression. DAI days after organ initiation. Scale bars, 500 μm (**a**, **b**), 100 μm (**c**–**f**). See also Supplementary Movie 6.

work highlights the importance of temporal and tissue-specific gradients that can act on different axes of growth but also in different yet overlapping time windows, thus providing a new foundation to explore mechanisms of spatial developmental modulation during multicellular development. Finally, our work has the potential to expand the repertoire of principles and strategies for the controllable regulation of morphogenesis through spatial-temporal finetuning of developmental gradients, thereby strengthening a foundation for applications in the emerging field of synthetic developmental biology[55,56].

## Methods
### Plant material and growth conditions
*pUBQ10::myr:YFP*[57], *pUBQ10::myr:TdTomato*[58], *pPIN1::PIN1:GFP*[59], *pPIN3::PIN3:GFP*[60], *pPIN7::PIN7:GFP*[61], *pYUC4::3xNLS-GFP*[39], *ap2-7*[45], and *spt-12*[62] were in Columbia background and *DR5v2::nls-3xVenus*[38] was in Columbia/Utrecht background. *crc-1*[42] in Landsberg *erecta* background was backcrossed to Columbia background (F5). *crc-1, spt-12*, and *pUBQ10::myr:TdTomato* were crossed and analyzed in F3. *ap2-7* mutant was crossed with *pUBQ10::myr:YFP* and analyzed in F3. *pUBQ10::myr:TdTomato* and *DR5v2::nls-3xVenus* were crossed and analyzed in F3. Plants were grown on soil in a growth chamber under long-day conditions (16 h illumination, 95 μmol m$^{-2}$ s$^{-1}$) with 60%–70% relative humidity at 22 ± 1 °C.

### Live-imaging microscopy
3-weeks-old flowers were dissected. Inflorescences were cut 2 cm below the apex, and the oldest floral buds were removed using fine tweezers to uncover young floral primordia. Floral buds at stage 7 and 8[33] were selected and initiating sepals, petals and stamens were manually removed with a needle and fine tweezers to expose the early gynoecium. Dissected floral buds were transferred to Ø60 mm petri dishes with 1/2 Murashige and Skoog (MS)[63] medium supplemented with vitamins, 1.5% agar (w/v), 1% sucrose (w/v), and 0.1% Plant Protective Medium (Plant Cell Technologies; v/v). Dissected stems were placed horizontally in a cavity cut out in the culture medium[64], immersed in water, and an abaxial gynoecium surface was imaged at 24 h intervals for up to 13 days. Between imaging, water was removed, and samples were transferred to a growth chamber with standard long day conditions (16 h illumination, 95 μmol m$^{-2}$ s$^{-1}$) with 60%–70% relative humidity at 22 ± 1 °C. The images shown in Figs. 1 and 2 are derived from two independent overlapping time-lapse experiments (2–4 DAI

and 4–13 DAI). The images shown in Fig. 4a, b are derived from two independent overlapping time-lapse experiments (2–5 DAI and 5–10 DAI). The images shown in Fig. 5e, f are derived from two independent overlapping time-lapse experiments (2-4 DAI and 4–10 DAI).

## Microscopy and image analysis

All the confocal imaging was performed using a Zeiss LSM 800 confocal laser scanning upright microscope with a 40× long-distance working, water-dipping objective (W Plan-Apochromat 40x/1.0 DIC VIS-IRM27). Excitation was achieved using a diode laser with 488 nm for YFP and GFP, and 561 nm for TdTomato. The emission was collected at 500–550 nm for GFP, at 490–520 nm for YFP/Venus, and at 600–660 nm for TdTomato. For each reporter line all images were acquired with the same microscopy settings. Confocal stacks were acquired using a step size of 0.5–1 μm distance in *z*-dimension, at 16 bits image depth, and 512 × 512 pixel resolution. For samples that were larger than the microscope field of view, multiple overlapping stacks were acquired and stitched using MorphoGraphX[34,65].

## Image analysis

The resulting confocal images were processed and analyzed using MorphoGraphX software[34,65]. Stacks were manually stitched into a single file using the "combine stack" tool. Surface detection was performed with the "edge detect" tool with a threshold from 8000–10,000 followed by "edge detect angle" (threshold 4000–6000). Initial meshes of 3–5 μm cube size were created and subdivided 2–3 times before projecting the membrane signal (2–4 μm from the mesh). Segmentation was performed with the "auto-segmentation" tool, followed by manual curation and segmentation of additional cells at the periphery of the mesh. Parent relations between cells of successive time points were attributed manually.

For lineage tracking analysis, the parent relations between each of the consecutive time points were combined to compute corresponding cell lineages over multiple days. The lineage data were later used to compute area expansion, cell growth anisotropy, and cell proliferation, and to trace stomata development. Area expansion is calculated as the relative increase between the surface area of a mother cell and the area of its daughter cell(s) at the next time point. It was expressed in percentage increase ([Total area of daughter cells at T1/Area of mother cell T0−1] ×100). Heat maps of growth, growth anisotropy, and cell divisions generated between consecutive time points were displayed in the second time point.

Growth rates along the longitudinal and mediolateral axis of gynoecium were calculated using a custom Bezier grid that was manually placed so that it closely followed the geometry of the organ at each time point. They were expressed as percentage increase ([Total length of daughter cells at T1/length of mother cell T0−1] ×100). Distance from the given region was calculated using the "Cell distance" tool, which measures the shortest path between cells following the organ surface geometry.

Reverse lineage tracing of cells corresponding to different organ regions (style, replum, and valves) was performed as described previously[64]. Briefly, we traced the origin of the different regions by following cell lineages assigned manually using the tool "Parent Label" from a later time point (13 DAI) when the style, replum and valves were clearly differentiated. Cells located in different regions were traced back, resulting in sectors that exclusively contributed to each tissue at 5 DAI.

Heat-maps of membrane localization of PIN proteins (*PIN1-GFP*, *PIN3-GFP*, *PIN7-GFP*) were performed as described previously[15]. GFP signal was projected from 2–4 μm from the extracted mesh. Arrows indicating PIN protein polarization were placed to enhance clarity and comprehensibility. The orientation of the arrows is based on enrichment of the signal at the specific membranes, U-shapes localization, and previously published results[24,40].

Images were assembled in Adobe Photoshop, Adobe Illustrator, and Microsoft PowerPoint. Regions (Fig. 1c) or cell types (Figs. 4e and 6c, f, Supplementary Fig. 7c) were selected and marked with specific labels (color) in MorphoGraphX software. For Fig. 1c, colored regions were displayed on the mesh with projected epidermal cell outlines. For stomata distributions only colored cells were kept while the remaining labels were deleted. Images were further processed in Adobe Photoshop by manually adding organ outlines and the median region.

The scripts for data analysis were written in R studio with R version 4.2.2.

## Chemical treatment

For 1-N-naphthylphthalamic acid (NPA), flowers were treated for three consecutive days 1 week after bolting with the solution containing 30 μM 1-N-naphthylphthalamic acid (Sigma) and 0.010 % (v/v) Silwet L-77. All treated plants were then grown under long-day conditions (16 h illumination, 95 μmol m$^{-2}$ s$^{-1}$) with 60%–70% relative humidity at 22 ± 1 °C. One week after the last NPA treatment, flowers were dissected for live-imaging that was performed without adding NPA.

## Reporting summary

Further information on research design is available in the Nature Portfolio Reporting Summary linked to this article.

## Data availability

All meshes and raw data underlying all reported averages are available to download from the Open Science Framework repository (https://osf.io/5p4xb/).

## Code availability

All R-scripts used to analyze data are available to download from the Open Science Framework repository (https://osf.io/2akhr/).

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

## Acknowledgements

We thank Olivier Hamant and Lars Østergaard for the critical reading of the manuscript. We thank Carlos Montoya for help with the sample segmentations, and Dolf Weijers, Zhengjuan Zhang, and Kristoffer Jonsson for seeds of reporter lines. This work was supported by a New Initiative grant from Centre SEVE to D.K. and Fonds de Recherche du Québec Nature et Technologies Team Grant (2021-PR-282285) to D.K. and A-L.R-K. D.K. research is funded by Discovery grants from the Natural Sciences and Engineering Research Council of Canada (RGPIN-2018-05762). A-L.R-K. research is funded by Discovery grants from the Natural Sciences and Engineering Research Council of Canada (RGPIN-2018-04897). E.B. was funded by the Natural Sciences and Engineering Research Council of Canada undergraduate study grant (BRPC-549453-2020). K.W. was funded by The Programa de Atracción de Talento 2017 (Comunidad de Madrid, 2021-5 A/BIO-20952), by Programa Estatal de Generación del Conocimiento y Fortalecimiento Científico y Tecnológico del Sistema de I + D + I PID2021-122158NB-I00 (2022-2025), and by the "Severo Ochoa Programme for Centres of Excellence in R&D" from the Agencia Estatal de Investigación of Spain (CEX2020-000999-S to the CBGP: 2022-2025). In the frame of SEV-2016-0672 and CEX2020-000999-S funding to CBGP, M.M. was supported with a postdoctoral contract. S.D.F. was supported by the Mexican National Council of Humanities, Science, and Technologies (CONAHCYT) grant CB-2017-2018-A1-S-10126.

## Author contributions

D.K. and A.G.F. conceived and designed the experiments; A.G.F. and H.B.R. performed all experiments with the help of B.W.; A.G.F., E.B., B.W., T.S., and J.B. extracted data from time-lapse series with the help from A-L.R-K.; A.G.F., B.W. M.M., and E.B. analyzed the data; B.W. generated movies; D.K. supervised the project; A.G.F., M.M., K.W., and D.K. wrote the paper with input from S.D.F. and A.-L.-R.-K.; D.K., A.-L.R.-K. and K.W. provided the funding. All the authors reviewed the paper.

## Competing interests

The authors declare no competing interests.
