## [Peer Review File · Nature Communications]

Two orthogonal differentiation gradients locally coordinate fruit morphogenesis.Reviewer #1 (Remarks to the Author):

The authors do a morphometrics based analysis of gynoecium formation in the model species *Arabidopsis*, tracking cellular parameters like cell growth and area extension (a proxy measure that accounts for cell division) across development. In addition to the patterns of growth the authors also infer patterns of differentiation (I am less clear on the basis for this, see below) and determine there are two orthogonal gradients of growth and differentiation in the gynoecium. One basipetal gradient in the style and a medio-lateral gradient in the carpel valves. Using computational modelling the authors hypothesize these gradients are controlled by two mobile signals that diffuse from the tip and from the carpel margin meristem, respectively. One of these is hypothesized to be auxin and some experimental validation via treatment with NPA is performed. Additional genetic tests are performed to validate the other signal.

The live-imaging of gynoecium development is extremely challenging from a technical point of view. The characterization of the cellular parameters of gynoecium development alone is an impressive achievement especially with the accompanying expression maps of auxin synthesis, transport and response maxima. Understandably, given the technical challenge, the authors have been selective with their experimental approach. This is not a problem in itself as there is a wealth of functional data on carpel morphogenesis including the roles of hormone signaling and organ identity specifying transcription factors. Unfortunately I don't feel that these have been properly integrated with the findings and modelling in this paper. I detail some of the issues below.

The main conclusions of the paper rest on a proposed mechanism for the regulation of two orthogonal gradients of growth and differentiation, and their role in whole organ morphogenesis. If I understood correctly the two gradients are found in the style and in the valves. The gradient in the style refers to non-homogenous cell growth in the longitudinal axis which is visible from 7DAI onwards. In the valve however there are two gradients in the medio-lateral axis; a gradient of cell growth, visible between 4-7DAI, and a gradient of cell differentiation, visible between 10-14DAI.

1. I am not sure I understand what role the very local and relatively late gradient of growth in the style plays in the model and in whole gynoecium development. Is there an hypothesized cross-talk between style and the valves? What would happen to the valves in the model if you remove that gradient from the style? Ultimately, does this gradient which appears to form de novo in the style have a role beyond style elongation?
2. It is not clear to me what evidence the authors are basing the existence of a medio-lateral differentiation gradient in the valves on, other than the appearance of stomata in the epidermis. Cell size and cell division rates, for example, seem to be evenly distributed within the valves across gynoecium development. This is important to clarify because the authors refer repeatedly to differentiation gradients being critical for patterning of the gynoecium.
3. It is also unclear whether and how the computational model takes into account the delay between growth and differentiation gradients in the valves and what mechanism is proposed for this. As far as I can tell the model only replicates growth gradients. Again this is important because the claim is that two competing differentiation gradients govern whole organ morphogenesis.
4. More importantly there is plenty of evidence in the literature that auxin plays a role in medio-lateral expansion of the valves and that this is dependent on polar auxin transport (for example, Larsson 2014). This causes two issues. First the effects of NPA treatment on gynoecium development are too broad to disentangle the role of the CMM signal in the medio-lateral from a longitudinal auxin signal.
5. Second, previous work also shows the importance of lateral auxin maxima and internal PAT for valve outgrowth, reporting on these dynamics in great detail. I understand this is technically challenging and what the authors have achieved here is already impressive, however I don't understand why they've chosen to not comment on the dynamics of auxin synthesis, transport and response maxima in the inner layers of the gynoecium when inferring and modelling auxin flow across the gynoecium. This is a real issue in my opinion. Despite the model doing a good job at

replicating the growth patterns and overall shape of the gynoecium, I am not convinced that it properly integrates our current understanding of hormone signaling dynamics in the gynoecium or that it is informative about the mechanisms that underlie morphogenesis in this organ.

Overall I think the originality and strength of this work is in the morphometric description of growth patterns in the gynoecium which is a great achievement and is a fundamentally important step in linking gene activity to organ morphogenesis. I hope that the authors will revisit their model, integrating these growth patterns with the known patterns of gene expression and hormone signaling.

Reviewer #2 (Remarks to the Author):

Review of Gomez-Felipe et al "Competing differentiation gradients...")

There are some interesting aspects of gynoecium development that are explored in this manuscript and the live-imaging approaches offer an exciting possibility for unique data sets and insights. The authors propose to study why the gynoecial valve domains (presumably an evolutionary modification of a leaf-like organ) display a different pattern of differentiation from Arabidopsis leaves. Arabidopsis leaves differentiate in a basi-petal fashion (from distal tip to base), where the valve domains appear to differentiate more uniformly along their longitudinal axis. This is differentiation pattern has been reported previously and is confirmed in this study. The authors find a pattern of basipetal differentiation present in the style domain. This is perhaps not previously reported in the literature.

The authors conclude that the medial domain prevents differentiation of the valve domains and propose that this is due to a diffusible signal that is generated by the medial domain and functions in the valve domains. The idea of a medial domain derived signal within the Arabidopsis gynoecium is not a new proposal as several groups have suggested this possibility previously, but the idea that this medial signal after diffusion inhibits the response of the valve cells to an apically derived (auxin?) signal and prevents differentiation is novel, to my knowledge. However, unfortunately this paper does not, to my satisfaction generate compelling evidence in favor of this mechanism over other potential mechanisms.

The weaknesses of the paper:

1) To me the model (or perhaps more properly the simulation) of gynoecial growth that is presented sets the assumptions that match the mechanisms that the authors set out to demonstrate. So to me the model/simulation indicates that such a system of patterning can result from the conditions set by the investigators, but there is not much evidence to suggest that the biological system actually works in this same fashion.

The ability of the model to recapitulate the effects of NPA are a bit of a stretch as I don't believe that auxin transport or synthesis is build into their model. The action of NPA in the model is defined as simply completely knocking out the activity of the medial domain derived signal.

2) The authors use the ap2 mutant to test the activity of the medial domain on differentiation. Their logic is if they place an ectopic medial domain on the sepal, they expect that the pattern of cellular differentiation in the sepal will be altered. Unfortunately, the ap2 mutant causes a dramatic and early identity specification defect, so the sepal is no longer a sepal. The sepal is largely converted at an early stage to a carpel. Thus, any changes to the pattern of differentiation might just and easily be explained by the organ identity mis-specification. For this approach to convince me, the authors would need to somehow cause ectopic generation of the medial domains on the sepal without converting the entire sepal to a carpel. If not, there is no way to disentangle the organ identity specification effect from the effect of the ectopic medial domain.

I believe that the Leunig ant double mutants strongly remove the medial derived tissues without

altering carpel identity, so this genetic combination might provide a system to look at the development of the carpel in the absence of a medial domain. I suspect that the differentiation of the Leunig ant double mutants carpels do not display a dramatic shift to a basipetal differentiation pattern, but I don't know of anyone who has looked. See "Regulation of Gynoecium Marginal Tissue Formation by LEUNIG and AINTEGUMENTA" Liu et al. *The Plant Cell*, Volume 12, Issue 10, October 2000, Pages 1879–1891. Unfortunately, this ant lug double mutant also has a dramatic effect on style formation, thus disrupting your proposed tip derived signal. You could look at the formation of stomata in the lug ant double mutant. If the medial domain is responsible for the mediolateral patterning of the valves (as evidenced by the mediolateral gradient of differentiation of stomata you have measured) then that gradient of differentiation should be disrupted in the lug ant double mutants.

I do believe that the live imaging and modeling approaches hold promise for providing insights to the development of this very complex structure, but in my opinion that the current version of this manuscript is not ready to be published in nature communications.

3) Perhaps most compelling developmental insight: line 138

Interestingly, the first signs of the establishment of the basipetal gradient of growth in the style (from 8 DAI) coincided with the elimination of the PIN1 expression in the epidermis (Fig.1a, Fig. 2d; Extended Data Fig. 3c). This suggests that acropetal (from the base to the tip) auxin transport in the style could help restrict auxin to its tip at early developmental stages to prevent precocious epidermal cell differentiation. At later stages, when PIN1 is eliminated from the epidermis, auxin could move through other PINs to more proximal regions triggering the basipetal gradient of cell differentiation.

Of course this is largely a correlation between the PIN patterns and differentiation patterns in the style, but an interesting correlation .

4) Another interesting observation:

I think the medio-lateral gradient of differentiation as measured by stomata development is interesting, but I wonder how different that is from leaf development. Leaves also grow from marginal meristems at the leaf edge and the leaf margin maintains a undifferentiated status longer than the non-marginal portions of the leaf. This could be due to a signal along the medial lateral axis, but could also be due to a margin specific activity that maintains a less differentiated state. This timing mechanisms, that invoke oldest cells differentiating first, might also explain the differentiation pattern along the mediolateral axis.

5) Some other aspects of the writing that are unclear and perhaps not well situated within a developmental biology perspective

There seems to be loose use of the terms "specify"/"specification" and "differentiation".

To the developmental biologist specification suggests the earliest stages of organ identity determination, that would typically be marked by organ identity genes being expressed and positional marker genes being expressed. For the gynoecium much of this happens at the earliest stages of gynoecial development (floral stage 5 the gynoecium is not yet morphologically distinct) and yet in the anlagen gene expression marks different portions of the gynoecial pre-primordia (See early work of John Bowman and expression patterns of yabby genes, CRC and AG and STM from various groups. Clearly the gynoecium is patterned (and I would argue specified) well before 4 DAI (perhaps 0 DAI is correct for specification).

Regarding differentiation as a term, I think that is what you more frequently are measuring. Even this has different stages (cessation of cell division, alteration in cell expansion, sometimes endoreduplication, and finally formation of final external cell morphology). So I think that the paper could be written more clearly with a careful attention to the terms "specify" and "differentiate" to clarify your arguments. Some examples are given below.

For example "line 86 "Interestingly, upon valve specification at 4 DAI,"

Valve specification happens well before 4 DAI. Specification of the valve domain and the differential fate specification of the medial and valve domains happens around stage 5 or 6 of floral development (likely even before the initiation of the gynoecium. During the anlagen stage, (about stage 5 of floral development) the expression of lateral (valve) domain markers and medial domain markers are spatially restricted. See Bowman and the expression of Yabby markers in the valve domain. (see "CRABS CLAW, a gene that regulates carpel and nectary development in Arabidopsis, encodes a novel protein with zinc finger and helix-loop-helix domains John L. Bowman[‡] and David R. Smyth*" for in situ of YABBY family member CRC in lateral domains as early as stage 6 (1DAI).

Line 180 – "CMM is the source of a putative mobile signal (CMMsig) that triggers the identity of the valve and inhibits differentiation in a concentration-dependent manner"

This is a misuse of the term identity (valve identity is specified much earlier). I think you just mean that differentiation is controlled by the signal.

Lines 95-97 – not very clear what you are trying to say here:

Our data suggest that time-shifted gradients of cellular behaviors are controlled independently along longitudinal and mediolateral axes in different parts of the developing gynoecium. Thus, two developmental gradients could locally tailor gynoecium morphogenesis along these perpendicular growth axes at different developmental windows

6) More here on my opinion of the model.

The model seems to be set up with parameters that are based on the suppositions of the authors, so the model suggests to me that what the authors propose could be true, but not that it is true. There are several other mechanistic explanations that might result in the same pattern. I suspect that the lateral domain derived tissues (valves) and the medial domain derived tissues (style and replum) are very different from very early stages of development and thus the valve domain cells may be non-responsive to any tip derived signal due to early changes in cell identity.

If I understand it correctly auxin is not put into the model.

In methods it states...

"To replicate NPA treatment (Fig. 4b-c) CMMsig is abolished from the model," So the built in assumption here is that NPA treatment deletes the CMM and the proposed signal, but no other effects of NPA – auxin transport are altered. This appears to be a weakness of the model.

Reviewer #3 (Remarks to the Author):

The manuscript by Gómez-Felipe et al., entitled 'Competing differentiation gradients coordinate fruit morphogenesis', reports a study combining quantitative live imaging with a mathematical model during Arabidopsis gynoecium development. The two-week live imaging at the single cell-level resolution revealed temporal and spatial gradients in cell growth, its anisotropy, cell divisions, and cell area. The authors found a sequential occurrence of the growth gradients in the mediolateral axis first and at the longitudinal axis second, and a gradient of cell differentiation in the mediolateral axis in later stages. Based on the observations, they hypothesized the competing differentiation gradients, as entitled. A mathematical model simulated the growth pattern. NPA treatment of wild-type gynoecium in vivo altered the growth patterns. A carpelized mutant sepal in vivo exhibited a mediolateral gradient of cell differentiation, similar to wild-type gynoecium. Overall, the time-shifted growth gradient in the orthogonal axes on a long time scale is an interesting observation, and the competing differentiation gradients is a potentially unique concept for plant organogenesis.

However, the manuscript includes multiple problems with the logical development and the model

rationales. The following issues hamper the assessment of how accurately the mathematical model and plant experiments prove the main conclusion of the manuscript, "directed by competition between orthogonal time-shifted differentiation gradients" (Line 272-3).

Main Comments

1. "we propose that it eventually outcompetes the latter longitudinal gradient by desensitizing valves to a tip-derived signal by triggering valve cell differentiation at early stages" (Line 166-8). The reviewer could not find the definition of the keywords "compete," "desensitizing," and "differentiation" in the mathematical model.
2. BP "signal appears later in the simulation corresponding to 7-11 DAI" (Line 535-6), whereas "CMMsig is deactivated" (Line 497) in phase #3 (7-12 DAI; Line 492-3). The descriptions expect these two signal gradients not to interact and not to work simultaneously in the model, which is somehow inconsistent with "competing" differentiation gradients.
3. "the computer model recapitulated general patterns of gynoecium growth that were comparable to that observed experimentally. Similarly to experimental data, mediolateral growth concentrated solely in the valves and the longitudinal gradient of growth was restricted to the style" (Line 185-8). The former sentence is an incomplete description of which "patterns" are "comparable" and to what degree, except for the latter sentence. In addition, the latter was similar to the following model assumptions; "we assumed that cells in the valves start to differentiate and slowly acquire isotropic growth, while cells in the replum and style stay undifferentiated and conserve the initial longitudinal elongation" (Line 486-8); "we introduced the basipetal gradient of growth in the style" (Line 493-4). Therefore, it remains to be seen what noteworthy findings and which part of the competing differential gradients theory this paragraph proves.
4. Regarding "experimentally-derived assumptions" (Line 170) of the model, the subsequent statements (Line 170-184 and 480-499) hardly specify which assumptions are derived from experimental facts (with citation of some references or figures) and which are matters to be proven. Moreover, the rationale for the model Equations (1)-(8) and parameters (Table 1) are little described from biological and physical points of view. For example, a model core representing "the competing gradients" is probably Equations (3) and (5), where the sum of two chemical concentrations (CMMsig and BPsig) with a gradient in orthogonal axes nonlinearly controls the rest length of each mesh in synergy with the strain (σ_e). However, the reviewer did not find the experimental evidence and theoretical backgrounds. Therefore, it is not easy to assess the reliability of this model and whether it explicitly proves the competing differentiation gradients despite many assumptions.
5. Some equations seem mathematically incorrect or inconsistent with relevant descriptions. If Equation (2) is true, the rest length E_e is always zero, contradicting Equation (3) describing E_e as a variable. The right-hand side of Equation (6) is always positive, expecting the temporal divergence of anisotropy A_f , which is inconsistent with the definition "range from zero to 1" (Line 519-20).
6. Experimental verification of CMM activity and its interpretation; "we removed CMM activity using naphthylphthalamic acid (NPA) treatment" (Line 216-7); "Cytokinin is particularly plausible candidate for such a signal as it is active in the CMM, can diffuse through the tissue" (Line 281-2). For Line 281-2, do the authors infer cytokinin is CMMsig? If true, how do the authors think NPA acts on cytokinin?
7. Figure 3g, 3i, 4j, 4l, Extended Data Fig. 3e: How did the authors determine whether the PIN polarization is basipetal or acropetal for each cell, indicated by arrows?

Minor Comments

Line 91-94: Which Figure does represent the "corresponding gradients of cellular differentiation revealed by the appearance of the specialized cells" in style? The reviewer wonders whether Extended Data Fig. 1b, referred to here, shows this property in the cell size of style. Figure 2f-g does not include style.

Line 143 - 144: Which Figure does show "basipetal gradient of cell differentiation"?

Line 173-5: "Phase#2 (4-7 DAI): new gynoeceum regions (valves, replum, and style) emerge, displaying divergent growth dynamics (fast and predominantly isotropic growth in the valves, slow and anisotropic growth in replum, fast and anisotropic growth in the style)." How did the mathematical model equations (2)-(6), parameters (Table 1), or other settings represent the region-specific growth?

Line 180-1: "CMM is the source of a putative mobile signal (CMMsig) that triggers the identity of the valve and inhibits differentiation in a concentration-dependent manner (Fig. 2f)". How did the authors define "identity" in the model? In addition, "CMM signal is deactivated" (Line 497) in phase #3 (7-12 DAI; Line 492-3), as pointed out above, whereas Figure 2f showed a result at 10-14DAI. The inconsistency between the model setting and referred experimental result would be resolved by referring to other experiments in phase #2.

Line 505: " T_v is the turgor pressure." Is " T_v " "T"?

Equation (3):")" was used twice.

Line 511: "e is the strain." Is "e" " σ_e "?

Line 517-24: Could the authors clarify a quantitative relationship between the anisotropy factor A_f in the model and the growth anisotropy in vivo (e.g., Fig. 1b)? Also, does the model represent the differentiation solely by the anisotropy A_f or other factors? Suppose it is solely defined by A_f , given the model assumption that A_f is independent of the signal concentration (CMMsig and BPsig) shown in Equation (6). This concentration independence is inconsistent with the following description: "CMM is the source of a putative mobile signal (CMMsig) that triggers the identity of the valve and inhibits differentiation in a concentration-dependent manner" (Line 180-1).

Line 532-3: " d_f is the distance of face f from producer (the replum in this case)." Does the distance represent only the mediolateral direction?

Line 538-9: " d_f is the distance of face f from producer (the top of the organ in this case)." Does the distance represent only the basipetal direction but not the mediolateral one? If true, strangely, a single symbol d_f represents the distance with a different meaning from the above case.

Response to reviewer comments for manuscript no. NCOMMS-23-02703-T:

“Competing differentiation gradients coordinate fruit morphogenesis.”

We would like to thank the reviewers for their thoughtful and constructive comments that helped us improve this manuscript. Below we provide detailed answers to all requests and suggestions.

Reviewer #1 (Remarks to the Author):

The authors do a morphometrics-based analysis of gynoecium formation in the model species *Arabidopsis*, tracking cellular parameters like cell growth and area extension (a proxy measure that accounts for cell division) across development. In addition to the patterns of growth the authors also infer patterns of differentiation (I am less clear on the basis for this, see below) and determine there are two orthogonal gradients of growth and differentiation in the gynoecium. One basipetal gradient in the style and a medio-lateral gradient in the carpel valves. Using computational modelling the authors hypothesize these gradients are controlled by two mobile signals that diffuse from the tip and from the carpel margin meristem, respectively. One of these is hypothesized to be auxin and some experimental validation via treatment with NPA is performed. Additional genetic tests are performed to validate the other signal.

The live imaging of gynoecium development is extremely challenging from a technical point of view. The characterization of the cellular parameters of gynoecium development alone is an impressive achievement, especially with the accompanying expression maps of auxin synthesis, transport, and response maxima. Understandably, given the technical challenge, the authors have been selective with their experimental approach. This is not a problem in itself as there is a wealth of functional data on carpel morphogenesis including the roles of hormone signaling and organ identity specifying transcription factors. Unfortunately, I don't feel that these have been properly integrated with the findings and modeling in this paper. I detail some of the issues below.

The main conclusions of the paper rest on a proposed mechanism for the regulation of two orthogonal gradients of growth and differentiation, and their role in whole organ morphogenesis. If I understood correctly the two gradients are found in the style and in the valves. The gradient in the style refers to non-homogenous cell growth in the longitudinal axis which is visible from 7 DAI onwards. In the valve however, there are two gradients in the mediolateral axis; a gradient of cell growth, visible between 4-7 DAI, and a gradient of cell differentiation, visible between 10-14 DAI.

1. I am not sure I understand what role the very local and relatively late gradient of growth in the style plays in the model and in whole gynoecium development. Is there any hypothesized cross-talk between style and the valves? What would happen to the valves in the model if you remove that gradient from the style? Ultimately, does this gradient which appears to form de novo in the style have a role beyond style elongation?

Response #1: Thank you very much for appreciating our work. Indeed, we propose a crosstalk between gradients in the valve and style. In the WT, the growth gradient in the style transiently spans to valve shoulders around 11 DAI, locally affecting cell expansion (Fig. 1a; Fig. 3h-l; Extended Data Fig. 2a; Extended Data Fig. 3d-e). This is now highlighted in the main text (lines 67-69 and 154-156). As suggested by the Reviewer, we now provide additional simulation in which the late growth gradient emanating from the style is removed. This abolished the transient growth increase in the valve shoulders (see new Extended Data Fig. 6). Contrarily, if we remove gynoecium mediolateral patterning in updated models, the valveless gynoecium conserves their responsiveness to the style derived signal and the influence of this basipetal gradient on the valve becomes stronger (Fig. 4a, Extended Data Fig. 7). Our results from NPA pretreatment (Fig. 4b-c) agree with this model prediction.

Interestingly, NGA3 and/or STY1 (genes involved in style identity) overexpression leads to partial conversion of the ovary into style tissue, an increase in growth of the valve tips (more pronounced shoulders), perturb longitudinal patterning of the fruit, and reduce plant fertility (Trigueros *et al.*, Plant Cell 2009), supporting possible crosstalk between gradients in the style and valves. Additionally, the modulation of auxin flow through the style may play a role in valve morphogenesis in other species from Brassicaceae, such as *Capsella*, where auxin is known to accumulate at the tip of the valves that form pronounced shoulders post-fertilization (Dong *et al.*, Curr. Biol. 2019). We added these points to the discussion in the main text (lines 324-327 and 329-333).

2. It is not clear to me what evidence the authors are basing the existence of a medio-lateral differentiation gradient in the valves on, other than the appearance of stomata in the epidermis. Cell size and cell division rates, for example, seem to be evenly distributed within the valves across gynoecium development. This is important to clarify because the authors refer repeatedly to differentiation gradients being critical for patterning of the gynoecium.

Response #2: Stomata have been extensively used to monitor progression of organ differentiation (e.g., Gonzalez *et al.*, Plant Cell 2012; Andriankaja *et al.*, Dev. Cell 2012; Kierzkowski *et al.*, Cell 2019; Silveira *et al.*, Plant Physiol. 2022; Le Gloanec *et al.*, Development 2022). Nonetheless, we fully agree with the Reviewer that additional cellular parameters can be used to monitor differentiation. We have now included the quantification of cell sizes in the valves and style, which clearly support the existence of the longitudinal and mediolateral differentiation gradients (see new Extended Data Fig. 1f-h).

3. It is also unclear whether and how the computational model takes into account the delay between growth and differentiation gradients in the valves and what mechanism is proposed for this. As far as I can tell the model only replicates growth gradients. Again this is important because the claim is that two competing differentiation gradients govern whole organ morphogenesis.

Response #3: In the initial version of the manuscript, we monitored stomata as a clear indicator of the mediolateral progression of cell differentiation and showed that stomata appear only at 10 DAI. We would like to clarify that it doesn't mean that cell differentiation only starts at this late stage. We now added maps of stomata lineages development showing that first precursors of stoma cells (small cells resulting from asymmetric cell divisions) appear as early as 7 DAI (see new Extended Data Fig. 1h). We also quantified the distribution of cell sizes showing that epidermal cells in the lateral location in the valves start to expand as early

as 6 DAI (see new Extended Data Fig. 1e). This data shows that both growth and differentiation gradients are established early during valve development and that they overlap.

The reviewer is correct, the model is based on dynamic mediolateral and longitudinal growth gradients. It is, however, well documented that cell differentiation in lateral organs is coupled with a progressive cessation of cell divisions, and a decrease in cellular growth rates (Donnelly *et al.*, *Dev. Biol.* 1999; Andriankaja *et al.*, *Dev. Cell* 2012; Hervieux *et al.*, *Curr. Biol.* 2016; Fox *et al.*, 2018; Kierzkowski *et al.*, *Cell* 2019; Harline & Roeder, *Plant Methods* 2023). Longitudinal gradients of growth and differentiation in the style have comparable dynamics to the leaf as they occur along the principal axis of organ elongation. The main difference is that they are established around 1 week later in the style (~ 9 DAI) as compared to leaf (Le Gloanec *et al.*, *Development* 2022). We and others have shown that gynoecium (and later fruit) in *Arabidopsis* continuously elongates during its development (Extended Data Fig. 2a-d; Eldridge *et al.*, *Development* 2016; Ripoll *et al.*, *PNAS* 2019). Thus, early mediolateral gradients in the valves are established within the existing tissue, perpendicularly to the main axis of its growth. The dominance of the longitudinal growth over the mediolateral growth, especially at later developmental stages, likely obstructs the growth decrease that we would normally expect to accompany the cell differentiation in the lateral locations of the valves.

Taking this into account, we better connected cell growth to cell differentiation in the revised model. Specifically, we integrated the parameter that describes the valve responsiveness to the longitudinal gradient in the style. By reducing this parameter, we approximate the progression of cellular differentiation in valves. In the model, this responsiveness to the longitudinal gradient of growth is therefore decreased in Phase#3 to model differentiation progression and gradient competition. We modified the text (lines 180-195) and methods accordingly to describe the improved version of the model.

4. More importantly there is plenty of evidence in the literature that auxin plays a role in medio-lateral expansion of the valves and that this is dependent on polar auxin transport (for example, Larsson 2014). This causes two issues. First the effects of NPA treatment on gynoecium development are too broad to disentangle the role of the CMM signal in the medio-lateral from a longitudinal auxin signal.

Response #4: We fully agree that polar auxin transport (PAT) is important for valve growth. That is why we do not use the NPA to perturb PAT during the development of the gynoecium but only to initially remove mediolateral patterning and to abolish valve initiation. As described in the method section, we purposely treated plants with NPA for the last time one week before dissecting flowers for our confocal imaging. We then followed only small gynoecium primordia which developed a valveless phenotype. During entire time-lapse experiments (for an additional 8 days) flowers were grown in the absence of NPA. Based on the proper polarization of PIN transporters from the beginning of our confocal observations, we assume that at the time when we start our imaging, PAT function is largely recovered in the valveless gynoecium. This experiment thus provided valuable information on how the dynamics of late longitudinal gradient is modified in the absence of early differentiation of the valves but with largely unperturbed PAT.

To avoid further confusion, we modified the main text (lines 214-216), method section (lines 434-435), and NPA-related figure legends to more explicitly highlight that this experiment was done after and not with the continuous NPA treatment.

5. Second, previous work also shows the importance of lateral auxin maxima and internal PAT for valve outgrowth, reporting on these dynamics in great detail. I understand this is technically challenging and what the authors have achieved here is already impressive, however I don't understand why they've chosen to not comment on the dynamics of auxin synthesis, transport and response maxima in the inner layers of the gynoecium when inferring and modelling auxin flow across the gynoecium. This is a real issue in my opinion. Despite the model doing a good job at replicating the growth patterns and overall shape of the gynoecium, I am not convinced that it properly integrates our current understanding of hormone signaling dynamics in the gynoecium or that it is informative about the mechanisms that underlie morphogenesis in this organ.

Overall I think the originality and strength of this work is in the morphometric description of growth patterns in the gynoecium which is a great achievement and is a fundamentally important step in linking gene activity to organ morphogenesis. I hope that the authors will revisit their model, integrating these growth patterns with the known patterns of gene expression and hormone signaling.

Response #5: Thank you very much for appreciating our work. We agree with the Reviewer that PAT and auxin signaling plays an important role in valves development. We added more discussion on that matter in the revised manuscript and cite the previous work as suggested (lines 336-341).

Regarding models, our conceptual simulations aren't done at the cellular level, therefore they don't integrate auxin transport mediated by PINs nor auxin signaling. Instead, we use experimental results on auxin and PINs to position tip-derived gradients in the simulations. To add molecular information on auxin transport and signaling would require much more complicated cellular models composed of thousands of cells and this is currently not feasible using existing modeling techniques. We are aware of these limitations since every model represents the proximation of very complex developmental programs. As in the case of other plant organ growth simulations (e.g., Runions *et al.*, *New Phytol.* 2017; Whitewoods *et al.*, *Science* 2020; Richardson *et al.*, *Science* 2021) our conceptual models represent a higher level of abstraction with the potential to guide the development of more complex cellular models in the future. We now discuss the limitations of the model in the revised Discussion section (lines 348-351).

Reviewer #2 (Remarks to the Author):

Review of Gomez-Felipe et al "Competing differentiation gradients...")

There are some interesting aspects of gynoecium development that are explored in this manuscript and the live-imaging approaches offer an exciting possibility for unique data sets and insights. The authors propose to study why the gynoecial valve domains (presumably an evolutionary modification of a leaf-like organ) display a different pattern of differentiation from *Arabidopsis* leaves. *Arabidopsis* leaves differentiate in a basi-petal fashion (from distal tip to base), where the valve domains appear to differentiate more uniformly along their longitudinal

axis. This is differentiation pattern has been reported previously and is confirmed in this study. The authors find a pattern of basipetal differentiation present in the style domain. This is perhaps not previously reported in the literature.

The authors conclude that the medial domain prevents differentiation of the valve domains and propose that this is due to a diffusible signal that is generated by the medial domain and functions in the valve domains. The idea of a medial domain derived signal within the Arabidopsis gynoecium is not a new proposal as several groups have suggested this possibility previously, but the idea that this medial signal after diffusion inhibits the response of the valve cells to an apically derived (auxin?) signal and prevents differentiation is novel, to my knowledge. However, unfortunately, this paper does not, to my satisfaction generate compelling evidence in favor of this mechanism over other potential mechanisms.

The weaknesses of the paper:

1) To me the model (or perhaps more properly the simulation) of gynoecial growth that is presented set the assumptions that match the mechanisms that the authors set out to demonstrate. So to me, the model/simulation indicates that such a system of patterning can result from the conditions set by the investigators, but there is not much evidence to suggest that the biological system actually works in this same fashion.

The ability of the model to recapitulate the effects of NPA is a bit of a stretch as I don't believe that auxin transport or synthesis is built into their model. The action of NPA in the model is defined as simply completely knocking out the activity of the medial domain-derived signal.

Response #6: The model, although relatively simple, allows us to conceptually test the contribution of gradients to shaping the gynoecium. At any timepoint of the simulation the organ shape results from the cumulative effect of growth gradients at previous steps, therefore imagining what should be the effect of superimposed growth gradients on organ shape is not intuitive. We do not state it is the only possible mechanism, but it is supported with our subsequent experiments. We argue that the simulations are a useful tool to test ideas in a more concrete manner and check if they are plausible.

Evidence for the putative gradient interactions comes from the analysis of differentiation patterns observed in our time-lapse experiments where we remove (Fig. 4), add (Fig. 5), or restrict (new Fig. 6) early mediolateral gradients (likely controlled by CMM signal).

Regarding the NPA model, we apologize if this has been confusing. We now refer to it as a removal of mediolateral gradient rather than an "NPA" simulation. The main idea was to test how tip-derived gradients would contribute to gynoecium patterning in the absence of the mediolateral gradient. Accordingly, this is implemented by the absence of mediolateral gradient and increased responsiveness to tip-derived signal (auxin) in Phase #3 in our revised models (Fig. 4a, Extended Data Fig. 7). We do not model explicitly auxin transport or auxin signaling, as this would be technically unfeasible with state-of-the-art modelling techniques, given such a large number of cells (see response #5 to Reviewer #1). We modified the main text (lines 180-195 and 207-213) and model description in the method's section to indicate why and how we do such simulations.

2) The authors use the ap2 mutant to test the activity of the medial domain on differentiation. Their logic is if they place an ectopic medial domain on the sepal, they expect that the pattern of cellular

differentiation in the sepal will be altered. Unfortunately, the *ap2* mutant causes a dramatic and early identity specification defect, so the sepal is no longer a sepal. The sepal is largely converted at an early stage to a carpel. Thus, any changes to the pattern of differentiation might just and easily be explained by the organ identity mis-specification. For this approach to convince me, the authors would need to somehow cause ectopic generation of the medial domains on the sepal without converting the entire sepal to a carpel. If not, there is no way to disentangle the organ identity specification effect from the effect of the ectopic medial domain.

I believe that the *Leunig ant* double mutants strongly remove the medial derived tissues without altering carpel identity, so this genetic combination might provide a system to look at the development of the carpel in the absence of a medial domain. I suspect that the differentiation of the *Leunig ant* double mutants carpels do not display a dramatic shift to a basipetal differentiation pattern, but I don't know of anyone who has looked. See "Regulation of Gynoecium Marginal Tissue Formation by *LEUNIG* and *AINTEGUMENTA*" Liu et al. *The Plant Cell*, Volume 12, Issue 10, October 2000, Pages 1879–1891. Unfortunately, this *ant lug* double mutant also has a dramatic effect on style formation, thus disrupting your proposed tip derived signal. You could look at the formation of stomata in the *lug ant* double mutant. If the medial domain is responsible for the mediolateral patterning of the valves (as evidenced by the mediolateral gradient of differentiation of stomata you have measured) then that gradient of differentiation should be disrupted in the *lug ant* double mutants.

I do believe that the live imaging and modeling approaches hold promise for providing insights to the development of this very complex structure, but in my opinion that the current version of this manuscript is not ready to be published in nature communications.

Response #7: We thank the Reviewer for these very valid comments. We are aware that introducing CMM into the sepal without severely disrupting its identity is likely impossible. *ap2-7* sepals still exhibit typical sepal characteristics such as giant cells and display similar growth behavior to the WT sepal during initiation suggesting that it is not just a carpel. However, we agree with the Reviewer that using a mutant that strongly reduce medial-derived tissues without altering carpal identity would be very useful. Unfortunately, the *lug/ant* double mutant is no longer available from the authors (Liu *et al.*, *Plant Cell* 2000), nor stock centers. We, however, followed the growth of the *crc/spt* double mutant which displays a split phenotype, with partially fused carpels lacking style and papilla, strongly reduced medial derived tissues including septum and ovules (Alvarez and Smyth, *Int. J. Plant Sci.* 2001). Spatial reduction of the CMM domain in this mutant allowed the expansion of the longitudinal gradient toward more proximal regions of the organ (see new Fig. 6, lines 282-299). This new data together with our former results supports the idea that removing (Fig. 4), adding (Fig. 5), or restricting (new Fig. 6) early mediolateral gradient of differentiation (likely controlled by CMM activity) modulate organ shape via the restriction of the longitudinal differentiation gradient (controlled by auxin).

3) Perhaps most compelling developmental insight: line 138 Interestingly, the first signs of the establishment of the basipetal gradient of growth in the style (from 8 DAI) coincided with the elimination of the PIN1 expression in the epidermis (Fig.1a, Fig. 2d; Extended Data Fig. 3c). This suggests that acropetal (from the base to the tip) auxin transport in the style could help restrict auxin to its tip at early developmental stages to prevent precocious epidermal cell differentiation.

At later stages, when PIN1 is eliminated from the epidermis, auxin could move through other PINs to more proximal regions triggering the basipetal gradient of cell differentiation.

Of course this is largely a correlation between the PIN patterns and differentiation patterns in the style, but an interesting correlation.

Response #8: We thank the Reviewer for appreciating our findings.

4) Another interesting observation:

I think the medio-lateral gradient of differentiation as measured by stomata development is interesting, but I wonder how different that is from leaf development. Leaves also grow from marginal meristems at the leaf edge and the leaf margin maintains an undifferentiated status longer than the non-marginal portions of the leaf. This could be due to a signal along the medial lateral axis, but could also be due to a margin specific activity that maintains a less differentiated state. This timing mechanisms, that invoke oldest cells differentiating first, might also explain the differentiation pattern along the mediolateral axis.

Response #9: Leaf margin plays an important morphogenetic role enabling the formation of the marginal protrusions such as serrations, lobes, and leaflets via the PAT (Bilsborough *et al.*, PNAS 2011, Kierzkowski *et al.*, Cell 2019). However, leaf blade growth is driven by the cell expansion and cell proliferation occurring within the developing blade (sometimes referred to as plate meristem), not at the leaf margin (e.g., Donnelly *et al.*, Dev. Biol. 1999; Fox *et al.*, PLOS Biol. 2018; Tsukaya, Plant Cell 2021; Le Gloanec *et al.*, Development 2022; Harline and Roeder Plant Methods 2023). Gradients of growth, divisions, cell sizes, cell lobeyness, and stomata formation clearly progress basipetally through the leaf blade. Importantly, the oldest cells in the leaf blade (as well as in the leaf margin) are also located in the distal portions of the organ (see Fig. 6 in Le Gloanec *et al.*, Development 2022). In that sense, valve regions located close to the CMM are rather equivalents of the leaf base (which is located close to the SAM) than the leaf margin.

5) Some other aspects of the writing that are unclear and perhaps not well situated within a developmental biology perspective

There seems to be loose use of the terms "specify"/"specification" and "differentiation".

To the developmental biologist specification suggests the earliest stages of organ identity determination, that would typically be marked by organ identity genes being expressed and positional marker genes being expressed. For the gynoecium much of this happens at the earliest stages of gynoecial development (floral stage 5 the gynoecium is not yet morphologically distinct) and yet in the anlagen gene expression marks different portions of the gynoecial pre-primordia (See early work of John Bowman and expression patterns of yabby genes, CRC and AG and STM from various groups. Clearly the gynoecium is patterned (and I would argue specified) well before 4 DAI (perhaps 0 DAI is correct for specification).

Regarding differentiation as a term, I think that is what you more frequently are measuring. Even this has different stages (cessation of cell division, alteration in cell expansion, sometimes endoreduplication, and finally formation of final external cell morphology). So I think that the paper

could be written more clearly with careful attention to the terms "specify" and "differentiate" to clarify your arguments. Some examples are given below.

For example "line 86 "Interestingly, upon valve specification at 4 DAI,"

Valve specification happens well before 4 DAI. Specification of the valve domain and the differential fate specification of the medial and valve domains happens around stage 5 or 6 of floral development (likely even before the initiation of the gynoecium. During the anlagen stage, (about stage 5 of floral development) the expression of lateral (valve) domain markers and medial domain markers are spatially restricted. See Bowman and the expression of Yabby markers in the valve domain. (see "CRABS CLAW, a gene that regulates carpel and nectary development in Arabidopsis, encodes a novel protein with zinc finger and helix-loop-helix domains John L. Bowman† and David R. Smyth*" for in situ of YABBY family member CRC in lateral domains as early as stage 6 (1DAI).

Response #10: Thank you for pointing this out. When using the term "valve specification at 4 DAI" we did not mean that it is genetically specified at this stage but rather starts to display differential growth behavior than other gyniecial regions. We now removed the term specification in the manuscript to avoid confusion.

We also thank the Reviewer for suggesting that we should more carefully use the term "differentiation" and that differentiation has different stages. We addressed that point in the manuscript. For example, we now added lineage tracing for stomata development highlighting that the process of differentiation doesn't only start when we observe final cell morphology (External Data Fig. 1h).

Line 180 – "CMM is the source of a putative mobile signal (CMMsig) that triggers the identity of the valve and inhibits differentiation in a concentration-dependent manner"

This is a misuse of the term identity (valve identity is specified much earlier). I think you just mean that differentiation is controlled by the signal.

Response #11: Yes, we mean that the CMM signal sets mediolateral gradients of growth and differentiation in the valves. We corrected the text accordingly (lines 164-167).

Lines 95-97 – not very clear what you are trying to say here:

Our data suggest that time-shifted gradients of cellular behaviors are controlled independently along longitudinal and mediolateral axes in different parts of the developing gynoecium. Thus, two developmental gradients could locally tailor gynoecium morphogenesis along these perpendicular growth axes at different developmental windows

Response #12: We now shorten this part to make it clear (lines 96-100).

6) More here on my opinion of the model.

The model seems to be set up with parameters that are based on the suppositions of the authors, so the model suggests to me that what the authors propose could be true, but not that it is true. There are several other mechanistic explanations that might result in the same pattern. I suspect that the lateral domain derived tissues (valves) and the medial domain derived tissues (style and

replum) are very different from very early stages of development and thus the valve domain cells may be non-responsive to any tip derived signal due to early changes in cell identity.

Response #13: Our model tests the hypothesis we propose based on experimental observations. We agree there may be other explanations but our data points to a presented plausible scenario that is further supported by our computer model simulations. Regarding the reduced responsiveness of valves to longitudinal gradient, it is now captured in the revised model and expressed as a valve responsiveness parameter. This parameter is decreased in Phase #3 of gynoecium development as the progression of valve differentiation is evident.

If I understand it correctly auxin is not put into the model.

In methods it states... "To replicate NPA treatment (Fig. 4b-c) CMMsig is abolished from the model," So the built in assumption here is that NPA treatment deletes the CMM and the proposed signal, but no other effects of NPA – auxin transport are altered. This appears to be a weakness of the model.

Response #14: We apologize for the confusion. Indeed, we do not model auxin transport (see Response #6). The Idea behind this simulation is to show how the tip-derived signal influence the tissue growth in the absence of mediolateral gradient.

Experimentally, NPA was used to remove the early mediolateral gradient of differentiation, but we assume that its effects on PAT are largely absent during our time-lapse experiments. As described in the method section, we purposely treated plants with NPA for the last time one week before dissecting flowers and followed organ growth on medium lacking NPA. We assume that PAT is largely recovered during our confocal observations as evidenced by proper PIN protein polarization (Fig. 4).

Reviewer #3 (Remarks to the Author):

The manuscript by Gómez-Felipe et al., entitled 'Competing differentiation gradients coordinate fruit morphogenesis', reports a study combining quantitative live imaging with a mathematical model during Arabidopsis gynoecium development. The two-week live imaging at the single cell-level resolution revealed temporal and spatial gradients in cell growth, its anisotropy, cell divisions, and cell area. The authors found a sequential occurrence of the growth gradients in the mediolateral axis first and at the longitudinal axis second, and a gradient of cell differentiation in the mediolateral axis in later stages. Based on the observations, they hypothesized the competing differentiation gradients, as entitled. A mathematical model simulated the growth pattern. NPA treatment of wild-type gynoecium in vivo altered the growth patterns. A carpelized mutant sepal in vivo exhibited a mediolateral gradient of cell differentiation, similar to wild-type gynoecium. Overall, the time-shifted growth gradient in the orthogonal axes on a long time scale is an interesting observation, and the competing differentiation gradients is a potentially unique concept for plant organogenesis.

However, the manuscript includes multiple problems with the logical development and the model rationales. The following issues hamper the assessment of how accurately the mathematical model and plant experiments prove the main conclusion of the manuscript, "directed by competition between orthogonal time-shifted differentiation gradients" (Line 272-3).

Main Comments

1. "we propose that it eventually outcompetes the latter longitudinal gradient by desensitizing valves to a tip-derived signal by triggering valve cell differentiation at early stages" (Line 166-8). The reviewer could not find the definition of the keywords "compete," "desensitizing," and "differentiation" in the mathematical model.

Response #15: These definitions describe experimental observations that were the base to set up the three modeled developmental stages. We now clarify this in the revised manuscript (lines 168-179). Experimental growth patterns based on differentiation events were integrated into the whole organ model to predict the final shape of the gynoecium. In the revised model we introduce a parameter controlling valve responsiveness to the longitudinal gradient which reflects the progression of valve differentiation. We avoid using terms desensitizing as we agree it is too vague. Interactions between gradients in the model is captured by valves having reduced responsiveness to the longitudinal gradient in the style (valve responsiveness parameter). We describe the improved models in the main text (lines 180-213) and method section.

2. BP "signal appears later in the simulation corresponding to 7-11 DAI" (Line 535-6), whereas "CMMsig is deactivated" (Line 497) in phase #3 (7-12 DAI; Line 492-3). The descriptions expect these two signal gradients not to interact and not to work simultaneously in the model, which is somehow inconsistent with "competing" differentiation gradients.

Response #16: Indeed, the style-associated longitudinal gradient is introduced in Phase #3 and coincides with progressive differentiations of valves, now reflected in our model by a parameter that defines the valve responsiveness to this gradient. Gradient interactions occur by a mechanism in which early mediolateral gradient limits the activity of later longitudinal gradient by controlling valve outgrowth and differentiation status (reducing valve responsiveness parameter).

3. "the computer model recapitulated general patterns of gynoecium growth that were comparable to that observed experimentally. Similarly, to experimental data, mediolateral growth concentrated solely in the valves and the longitudinal gradient of growth was restricted to the style" (Line 185-8). The former sentence is an incomplete description of which "patterns" are "comparable" and to what degree, except for the latter sentence. In addition, the latter was similar to the following model assumptions; "we assumed that cells in the valves start to differentiate and slowly acquire isotropic growth, while cells in the replum and style stay undifferentiated and conserve the initial longitudinal elongation" (Line 486-8); "we introduced the basipetal gradient of growth in the style" (Line 493-4). Therefore, it remains to be seen what noteworthy findings and which part of the competing differential gradients theory this paragraph proves.

Response #17: We thank the reviewer for this comment. The idea behind gradient interactions involves the valve responsiveness to the longitudinal gradient in a style that enters Phase #3. In the revised model, it is controlled by the responsiveness parameter and biologically it relates to the reduction of valve responsiveness due to the progression of valve differentiation. This competition mechanism presumably precludes invasion of the longitudinal gradient to valve regions.

4. Regarding "experimentally-derived assumptions" (Line 170) of the model, the subsequent statements (Line 170-184 and 480-499) hardly specify which assumptions are derived from experimental facts (with citation of some references or figures) and which are matters to be proven. Moreover, the rationale for the model Equations (1)-(8) and parameters (Table 1) are little described from biological and physical points of view. For example, a model core representing "the competing gradients" is probably Equations (3) and (5), where the sum of two chemical concentrations (CMMsig and BPsig) with a gradient in orthogonal axes nonlinearly controls the rest length of each mesh in synergy with the strain (σ_e). However, the reviewer did not find the experimental evidence and theoretical backgrounds. Therefore, it is not easy to assess the reliability of this model and whether it explicitly proves the competing differentiation gradients despite many assumptions.

Response #18: In the revised manuscript we improved the overall description of our model. It is important to mention our models are whole organ models that are used to test the proposed concept. They currently do not include individual cells or processes modelled at cellular level such as auxin transport, PIN polarity or gene transcription. Instead, we focus on modeling temporally controlled gradients and their impact on overall organ development. As such CMMsig and BPsig are modeled as gradients of growth rate which are set based on qualitative comparison to experimental data (see revised Model description). Generally, we implement growth by relaxing walls. While the strain is modeled based on anisotropy patterns recorded in our time-lapse experiments.

We apologize that it may have been unclear. We have improved the model description in the revised manuscript and listed the detailed assumptions as sections of model description for clarity (Lines 435-568).

5. Some equations seem mathematically incorrect or inconsistent with relevant descriptions. If Equation (2) is true, the rest length E_e is always zero, contradicting Equation (3) describing E_e as a variable. The right-hand side of Equation (6) is always positive, expecting the temporal divergence of anisotropy A_f , which is inconsistent with the definition "range from zero to 1" (Line 519-20).

Response #19: We thank the reviewer for pointing out this mistake. We have corrected the Model description (Section 1, Growth implementation) as follows:

"Equation (2) is true, the rest length E_e is always zero, contradicting Equation (3) describing E_e as a variable."

We have replaced:

$$R_e = R_e \cdot (1 + E_e)$$

with, Equation 2

$$\frac{dR_e}{dt} = \alpha_e \cdot E_e$$

R_e is the current rest length of edge e ; α_e is the rest length update rate for edge e that is given by Equation 3

"The right-hand side of Equation (6) is always positive, expecting the temporal divergence of anisotropy A_f , which is inconsistent with the definition "range from zero to 1" (Line 519-20)."

We apologize for the typo, there was a missing negative sign in the original equation, but now it is corrected in Equation 9 (see Model Description, Section 4 Growth Anisotropy)

6. Experimental verification of CMM activity and its interpretation; "we removed CMM activity using naphthylphthalamic acid (NPA) treatment" (Line 216-7); "Cytokinin is particularly plausible candidate for such a signal as it is active in the CMM, can diffuse through the tissue" (Line 281-2). For Line 281-2, do the authors infer cytokinin is CMMsig? If true, how do the authors think NPA acts on cytokinin?

Response #20: Yes, as stated in the discussion, we think that cytokinin is one of the plausible candidates for the CMMsig as it can diffuse through the tissue and is known to delay cell differentiation. Cytokinin signaling monitored with TCS reporter occurs in the medial regions and seems to have mutually exclusive patterns with auxin during the gynoecium development (Marsch-Martinez *et al.*, Plant J. 2012). Upon NPA treatment (performed in a similar way as in our studies), cytokinin signaling increases all over the central region of the radialized gynoecium (Zuñiga-Mayo *et al.*, Front. Plant Sci. 2014). Such cytokinin response may be the reason why cells in the radialized gynoecium don't differentiate until they receive the tip derived signal (auxin). We now add this speculation in the main text (lines 336-341).

7. Figure 3g, 3i, 4j, 4l, Extended Data Fig. 3e: How did the authors determine whether the PIN polarization is basipetal or acropetal for each cell, indicated by arrows?

Response #21: The orientation of arrows is based on our quantifications with MorphoGraphX (i.e., preferential localization of PIN proteins on the wall oriented perpendicularly to the main axis of the organ, U-shapes expression pattern at the membrane) as well as on published literature showing the orientation of PINs polarisation at some stages of gynoecium development (e.g., Moubayidin & Ostergaard Curr. Biol. 2014; Larsson *et al.*, Plant Cell 2014). Furthermore, a similar approach was used in a substantial body of recent research (Heisler *et al.*, Curr. Biol. 2005; Barbier de Reuille *et al.*, eLife 2005; Bayer *et al.*, Genes Dev. 2009; Feraru *et al.*, Curr. Biol. 2011; Abley *et al.*, Development 2013; Tanaka *et al.*, PLOS Biol. 2013; Larsson *et al.*, Plant Cell 2014; Abley *et al.*, eLife 2016; Richardson *et al.*, Plant Cell 2016; Ditengou *et al.*, New Phytol. 2017; Burian *et al.*, Nat. Plants 2022).

We have incorporated these arrows to enhance the clarity and comprehensibility of our results. In the revised version of the manuscript, we have removed some arrows in cases where polarization may be less evident. We have also provided a clearer explanation of the purpose behind the placement of these arrows in the revised version of the manuscript (lines 425-427).

Minor Comments

Line 91-94: Which Figure represents the "corresponding gradients of cellular differentiation revealed by the appearance of the specialized cells" in style? The reviewer wonders whether

Extended Data Fig. 1b, referred to here, shows this property in the cell size of style. Figure 2f-g does not include style.

Response #22: We indeed refer to cell size increasing from the tip to the base (Extended Data Fig. 1b). We have now included the quantification of cell size in the style to better show this progression (new Extended Data Fig. 1e).

Line 143 - 144: Which Figure does show "basipetal gradient of cell differentiation"?

Response #23: Extended Data Fig. 1b and new Extended Data Fig. 1e.

Line 173-5: "Phase#2 (4-7 DAI): new gynoecium regions (valves, replum, and style) emerge, displaying divergent growth dynamics (fast and predominantly isotropic growth in the valves, slow and anisotropic growth in replum, fast and anisotropic growth in the style)." How did the mathematical model equations (2)-(6), parameters (Table 1), or other settings represent the region-specific growth?

Response #24: Tissue-specific parameters during different developmental phases as provided in the Model Description were indeed unclear, we apologize for this confusion. We substantially rewrote the model description section to state numbers for the regional growth rates and associated with Equation 4 and Section 2 in the Model description (Regional growth regulation). Model parameters with corresponding values are listed in Table 1.

Line 180-1: "CMM is the source of a putative mobile signal (CMMsig) that triggers the identity of the valve and inhibits differentiation in a concentration-dependent manner (Fig. 2f)". How did the authors define "identity" in the model? In addition, "CMM signal is deactivated" (Line 497) in phase #3 (7-12 DAI; Line 492-3), as pointed out above, whereas Figure 2f showed a result at 10-14DAI. The inconsistency between the model setting and referred experimental result would be resolved by referring to other experiments in phase #2.

Response #25:

How did the authors define "identity" in the model?

The valve, replum, and style regions defined in the model are now clearly stated in the Model Description section, these regions are associated with different growth dynamics and they appear in Phase #2. In this Phase, each location of the organ is assigned a "type" (style, valve, or replum). The mediolateral growth gradient that is entering #Phase 2, gradually decays at the beginning of Phase #3. The mediolateral gradient affects the valve responsiveness parameter that is successively reduced to reflect the progression of valve differentiation in Phase #3. Generally, we improved overall discussions on this matter in the revised manuscript (lines 188-191 and 210-212).

Line 505: " T_v is the turgor pressure." Is " T_v " "T"?

Response 26:

We thank you for spotting this error. We replaced:

T_v is the turgor pressure force

with

T is the turgor pressure force

Equation (3):" was used twice.

Response #27:

We removed the additional “)”

Line 511: "e is the strain." Is "e" " σ_e "?

Response #28:

e corresponds to the given edge. The strain component has been directly moved to Equation 4 for clarity.

Line 517-24: Could the authors clarify a quantitative relationship between the anisotropy factor A_f in the model and the growth anisotropy in vivo (e.g., Fig. 1b)? Also, does the model represent the differentiation solely by the anisotropy A_f or other factors? Suppose it is solely defined by A_f , given the model assumption that A_f is independent of the signal concentration (CMMsig and BPsig) shown in Equation (6). This concentration independence is inconsistent with the following description: "CMM is the source of a putative mobile signal (CMMsig) that triggers the identity of the valve and inhibits differentiation in a concentration-dependent manner" (Line 180-1).

Response #28:

Could the authors clarify a quantitative relationship between the anisotropy factor A_f in the model and the growth anisotropy in vivo (e.g., Fig. 1b)?

In the model, the anisotropy factor regulates the stiffness of the strain constraint for the PBD physics engine described later in section 8 and cited in (Müller *et al.*, Symp. Comp. Anim. 2014).

The anisotropy factor ranges from 0 (isotropic growth) to 1 (anisotropic growth).

The experimentally observed tissue anisotropy (Fig. 1b) is defined as the ratio of expansion in the maximal and minimal principal growth directions, following anisotropy is represented by a maximal anisotropy factor (1).

Also, does the model represent the differentiation solely by the anisotropy A_f or other factors?

The improved model includes growth gradient profiles and independently the anisotropism of growth that are closely related to those observed experimentally. To connect growth to differentiation we used the parameter theta (see revised Model description, Sections 1 & 2 Growth Implementation and Regional growth description). This parameter describes the responsiveness of valves to the longitudinal gradient in Phase #3. Specifically, it is reduced to reflect the progression of valve differentiation. We improved the description of the Model in the revised version of the manuscript. The growth anisotropy section has been included in the revision and described the details of Anisotropy regulation in the model. Anisotropy does not represent differentiation but growth orientation solely.

Line 532-3: "d_f is the distance of face f from producer (the replum in this case)." Does the distance represent only the mediolateral direction?

Response #29:

We apologize for not being clear about this point. The distance D (Equations 5 & 7) represents the "surface distance", which is the sum of the distances between faces centroid along the shortest path from the current face and the source of the signal. The shortest path was calculated using the Dijkstra algorithm; we have included the description of this process in the revised Methods (Model Description, Section 3, Growth gradient definition).

Line 538-9: "d_f is the distance of face f from producer (the top of the organ in this case)." Does the distance represent only the basipetal direction but not the mediolateral one? If true, strangely, a single symbol d_f represents the distance with a different meaning from the above case.

Response #30:

Please see former response #29.

Reviewer #1 (Remarks to the Author):

Dear authors

I have read your replies to the reviewers and unfortunately, I do not think that our concerns have been fully addressed.

I note that, while we agreed that the live-imaging and morphometrics work done here is important and provides novel insight, we were also unanimous in our concerns about the weaknesses of the manuscript and specifically about the limitations of the model as it was presented. In summary, a major concern was that there was no clear evidence for the interaction (competition in the original manuscript) of the two gradients and that the model was biased to show this mechanism could generate the observed growth patterns but not to fully test different hypotheses for how this could be controlled. Unfortunately, despite the revision from "competing gradients" to "modulation of gradients" I don't think these issues have been addressed in the responses or the revision of the manuscript.

My specific concern was for the lack of clear mechanism for how the longitudinal gradient which appears to be restricted to the style and operate at a different time could influence valve development. In the revised manuscript you've revised the "restriction" of the longitudinal gradient and propose that this could "invade" the valves. Although I appreciate the discussion of the literature on the overexpression of style identity genes to support a crosstalk between style and valves, I find the evidence that this manuscript provides for a mechanism for this crosstalk tenuous. The invasion of the longitudinal gradient to the valves is still very limited both in area and time therefore it is unclear to me how this could have a major effect in overall growth and differentiation of the carpel. It is also difficult to reconcile how this continuous gradient from the style can "invade" the top of the valves at 11DAI when there is a very clear visible boundary between style and valves which is marked by a low area of expansion.

The addition of a new parameter to force the responsiveness of the valves to a signal from the style is also problematic, in my view, since it inputs into the model the very assumption for which the model is supposed to be providing a test. As other reviewers point out in the earlier reviews, it is unclear how the can model support the theory put forward by the authors since it doesn't accurately provide a test of competing hypotheses. Critically, is an interaction between the gradient in the style and valves necessary to explain the observed growth and differentiation patterns or could there be an alternative mechanism?

As I said in the earlier review, the live-imaging and morphometrics description of carpel growth are an important result (the addition of the analysis of *crc spt* mutant is really interesting too!) and provide novel insight into development of this structure as well as great promise for our understanding of how these processes are controlled. However, the model as it is presented has a limited use as it does not provide a real test for the hypotheses put forward.

Reviewer #2 (Remarks to the Author):

This version of the manuscript has addressed some of the concerns and points I previously raised. However, I still have significant concerns. These concerns that I detail below, in my opinion, reduce the significance of the work. I feel that the authors have overstated the significance and certitude of their conclusions given the data presented. Although there are items of value in this manuscript, I believe that the manuscript's value is diminished by overly bold conclusions that are not fully supported.

Additionally, as presented it is difficult to separate what is novel in this manuscript from ideas and data presented in previous work. This work would be stronger if presented in better context to existing studies. For example, both gynoecial apical-basal and medio-lateral gradients have been previously proposed and auxin and cytokinin have been postulated as parts of these gradients. What would be new here is the interaction between the two gradients, but I find compelling evidence of that interaction lacking.

I am unaware of other data demonstrating the relatively late (developmentally) gradient of differentiation within the valve domain that the authors show that affects both cell growth and stomatal differentiation. This to me is one of the bright spots of the manuscript. More effort could be put into better defining the mechanisms of that patterning event.

The other bright point has to do with the patterning and cell growth characteristic within the style. However, I feel that the entire effort to connect the two patterning events is not well supported. The experiments that the authors carry out to alter the medial domain all are confounded with concomitant changes to organ identity. This is true in both *ap2* mutants that convert sepals to carpels and in the *spt crc* double mutant that is believed to reduce the carpel identity in the whorl four organs, as well as the NPA treatments.

I empathize with that authors as what they need is a loss-of-function or a gain of function manipulation that alters the medial/marginal domain without altering carpel identity. I do not believe that they have that currently. The in *ap2* mutants there can be a variety of phenotypes exhibited these are variable in their strength both from organ to organ and also based on the specific allele. However, what the authors show in figure 4e and 4f represents a strong conversion of the sepal to a carpel. The shape of the chimeric organ reveals the strong carpel identity. The authors indicate that there are some sepal giant cells, but from my limited ability to assay that in this image, there seem to be few. Past work suggests that the organ is mostly converted to carpel with small patches of sepal identity perhaps retained. AG is expected to be derepressed in these organs from an early stage and drives carpel specification, along with CRC.

relevant references:

CRABS CLAW and SPATULA, two Arabidopsis genes that control carpel development in parallel with AGAMOUS - John Alvarez, David R. Smyth

Drews, G. N., Bowman, J. L. and Meyerowitz, E. M. (1991). Negative regulation of the Arabidopsis homeotic gene AGAMOUS by the APETALA2 product. *Cell* 65, 991-1002

And quoting from "Control of Arabidopsis Flower and Seed Development by the Homeotic Gene APETALA2" K. Diane Jofuku, et al.

In weak, partial loss-of-function *ap2* mutants, sepals are homeotically transformed into leaves, and petals are transformed into pollen-producing stamenoid organs (Bowman et al., 1989, 1991b). By contrast, in strong *ap2* mutants, sepals are transformed into ovule-bearing carpels, petal development is suppressed, the number of stamens is reduced, and carpel fusion is often defective (Komaki et al., 1988; Kunst et al., 1989; Bowman et al., 1991b)

Thus, I think that the figure 5 title "Introducing marginal meristem activity in the sepal largely eliminates the longitudinal developmental gradient" is misleading. The authors are not simply "introducing marginal meristem activity" they are creating a largely carpelloid organ that appears to follow the developmental trajectory that carpels typically follow.

Figure 4 is also challenging to interpret. NPA treatment is also variable and depends on the strength of the treatment. The organs shown in figure 4 display very strong NPA treatment phenotypes. These organs appear to be entirely radialized and entirely lacking valve domains. Previous authors have suggested that these radialized organs represent an organ without ovary domains. The identity of the remaining organ has been hypothesized to be style apically and stipe/or gynophore basally. Patterns of internal vascularization support this. (Auxin and ETTIN in Arabidopsis gynoecium morphogenesis. - Jennifer L. Nemhauser, Lewis J. Feldman, Patricia C. Zambryski or "A model for an early role of auxin in Arabidopsis gynoecium morphogenesis" - Charles Hawkins and Zhongchi Liu, 2014 and "Polar auxin transport together with AINTEGUMENTA and REVOLUTA coordinate early Arabidopsis gynoecium development" Nole-Wilson et al. 2010)

Although the authors document the spreading of the auxin gradient into more basal locations upon NPA treatment, the role of the mediolateral gradient in this process is unclear in this experiment as

well. Given the previous observations suggesting an expansion of the length of the style domain upon NPA treatment, one might expect that auxin responses would expand basally due to the transformation of tissue identity. Thus, I find the statement in Line 232 to be overly speculative. (i.e. "These results support the idea that an early differentiation in the valves along the mediolateral axis (likely controlled by CMM signal) is a key event to limit the tip-derived signal from setting up an organ-wide basipetal gradient of cell differentiation.")

I applaud the authors for adding the *crc* spt experiment, but I don't think that it adds much as *crc* is thought to control AG-independent aspects of carpel identity and so you still have a confounding of carpel identity with medial domain.

Line 270 is overly speculative: (i.e. "These findings confirm that the CMM-dependent, early establishment of mediolateral gradient prevents the progression of late basipetal cell differentiation throughout the entire organ.")

Similarly, the abstract is overly speculative, in my opinion.

I am being rather strict and harsh in my assessment. That is in part driven by the high caliber of the Nature Communications standards. It is also driven by my concerns that the data is over interpreted. I think the authors will have success getting this work published, perhaps in a different journal. I think that the paper should be rewritten more-carefully. This will perhaps mitigate the desired impact of the paper, but I don't feel that it is appropriate to make the bold conclusions that are being made here.

One unanswered question that arises from the model:

If there is an interaction between the apical/basal gradient and differentiation in the valve, why is the rate of differentiation between the top of the valve and the bottom of the valve similar. Shouldn't the cells at the bottom of the valve domain act differently from those at the top of the valve. Unless you propose a threshold response effect.

Minor point –

I don't believe that the data in Extended Data figure 4 is representative of what is happening. If you were to focus up and down through the gynoecium to count the initiating ovule primordia, you would see that almost all the ovule primordia arise nearly simultaneously. There appears to be a miscounting in your approach using just the digital longitudinal section. I believe several groups have reported that the ovule initiation is nearly simultaneous in Arabidopsis. I don't have reference here, but I believe it is work of Chuck Gasser and perhaps Robinson-Beers or Kai Schneitz. You can count them more accurately with chloral hydrate cleared gynoecia.

Response to reviewer comments.

We would like to thank the reviewers for their additional feedback. Below we provide detailed answers to all requests and suggestions.

Reviewer #1 (Remarks to the Author):

Dear authors

I have read your replies to the reviewers and unfortunately, I do not think that our concerns have been fully addressed.

I note that, while we agreed that the live-imaging and morphometrics work done here is important and provides novel insight, we were also unanimous in our concerns about the weaknesses of the manuscript and specifically about the limitations of the model as it was presented. In summary, a major concern was that there was no clear evidence for the interaction (competition in the original manuscript) of the two gradients and that the model was biased to show this mechanism could generate the observed growth patterns but not to fully test different hypotheses for how this could be controlled. Unfortunately, despite the revision from “competing gradients” to “modulation of gradients” I don’t think these issues have been addressed in the responses or the revision of the manuscript.

My specific concern was for the lack of clear mechanism for how the longitudinal gradient which appears to be restricted to the style and operate at a different time could influence valve development. In the revised manuscript you’ve revised the “restriction” of the longitudinal gradient and propose that this could “invade” the valves. Although I appreciate the discussion of the literature on the overexpression of style identity genes to support a crosstalk between style and valves, I find the evidence that this manuscript provides for a mechanism for this crosstalk tenuous. The invasion of the longitudinal gradient to the valves is still very limited both in area and time therefore it is unclear to me how this could have a major effect in overall growth and differentiation of the carpel. It is also difficult to reconcile how this continuous gradient from the style can “invade” the top of the valves at 11 DAI when there is a very clear visible boundary between style and valves which is marked by a low area of expansion.

The addition of a new parameter to force the responsiveness of the valves to a signal from the style is also problematic, in my view, since it inputs into the model the very assumption for which the model is supposed to be providing a test. As other reviewers point out in the earlier reviews, it is unclear how the can model support the theory put forward by the authors since it doesn’t accurately provide a test of competing hypotheses. Critically, is an interaction between the gradient in the style and valves necessary to explain the observed growth and differentiation patterns or could there be an alternative mechanism?

As I said in the earlier review, the live-imaging and morphometrics description of carpel growth are an important result (the addition of the analysis of *crc* spt mutant is really interesting too!) and provide novel insight into development of this structure as well as great promise for our understanding of how these processes are controlled. However,

the model as it is presented has a limited use as it does not provide a real test for the hypotheses put forward.

Response #1:

Thank you very much for all your feedback. We agree that the idea of gradient competition cannot be sufficiently supported by the experiments we could perform. For example, due to the experimental limitation, we cannot remove or introduce the CMM activity without perturbing patterning or identity of the tissues composing the gynoecium. We also agree that the simulations as presented could be perceived as being biased toward the support of the interactions between orthogonal gradients.

We now refocused the manuscript on a morphometrics description of gynoecium development based on our live-imaging data. In contrast to previous work, where gynoecium morphogenesis was suggested to be controlled at the organ level by global polarity fields spanning the entire epidermis (Eldridge et al., 2016), we now propose an alternative mechanism where orthogonal differentiation gradients act locally and are largely dependent on tissue identity. Importantly, sequential introduction of differentiation gradients (first in the valves and then in the style), seems to ensure simultaneous maturation of ovules and thus reproductive success.

We also decided to remove the computational model of the gradient competition. The idea that two gradients could influence each other is now mentioned only in the discussion (suggested by some of the data but requiring future investigation).

Reviewer #2 (Remarks to the Author):

This version of the manuscript has addressed some of the concerns and points I previously raised. However, I still have significant concerns. These concerns that I detail below, in my opinion, reduce the significance of the work. I feel that the authors have overstated the significance and certitude of their conclusions given the data presented. Although there are items of value in this manuscript, I believe that the manuscript's value is diminished by overly bold conclusions that are not fully supported.

Response #2:

We apologize for our bold conclusions and now refocus our manuscript on clear correlations between tissue identities and the newly uncovered orthogonal gradients emerging in various tissue types. We believe that by doing so our conclusions are now strongly supported by our biological data.

Additionally, as presented it is difficult to separate what is novel in this manuscript from ideas and data presented in previous work. This work would be stronger if presented in better context to existing studies. For example, both gynoecial apical-basal and medio-lateral gradients have been previously proposed and auxin and cytokinin have

been postulated as parts of these gradients. What would be new here is the interaction between the two gradients, but I find compelling evidence of that interaction lacking.

Response #3:

We are sorry for this confusion. In the new version of the manuscript, we provide an additional literature review including the role of auxin in mediolateral and longitudinal patterning (see the introduction). We hope that this helps to clearly discriminate between previously reported mechanisms of gynoecium patterning from newly reported tissue-specific growth and differentiation dynamics that we propose to be crucial for its specialized function.

I am unaware of other data demonstrating the relatively late (developmentally) gradient of differentiation within the valve domain that the authors show that affects both cell growth and stomatal differentiation. This to me is one of the bright spots of the manuscript. More effort could be put into better defining the mechanisms of that patterning event.

The other bright point has to do with the patterning and cell growth characteristic within the style. However, I feel that the entire effort to connect the two patterning events is not well supported. The experiments that the authors carry out to alter the medial domain all are confounded with concomitant changes to organ identity. This is true in both *ap2* mutants that convert sepals to carpels and in the *spt crc* double mutant that is believed to reduce the carpel identity in the whorl four organs, as well as the NPA treatments.

Response #4:

Thank you very much for appreciating our data. In the new version of the manuscript, we discuss further how the mediolateral gradient could be controlled. For example, we discussed the role of auxin as a potential player in the regulation of this gradient apart from a CMM derived signal. Unfortunately, testing the role of CMM in controlling the mediolateral gradient is experimentally very challenging as it is impossible to remove or introduce the CMM activity without perturbing the identity of the organ. We, therefore, refocused the manuscript on the role of time-shifted gradients that are tissue-dependent and controlled locally.

I empathize with that authors as what they need is a loss-of-function or a gain of function manipulation that alters the medial/marginal domain without altering carpel identity. I do not believe that they have that currently. The *in ap2* mutants there can be a variety of phenotypes exhibited these are variable in their strength both from organ to organ and also based on the specific allele. However, what the authors show in figure 4e and 4f represents a strong conversion of the sepal to a carpel. The shape of the chimeric organ reveals the strong carpel identity. The authors indicate that there are some sepal giant cells, but from my limited ability to assay that in this image, there seem to be few. Past work suggests that the organ is mostly converted to carpel with

small patches of sepal identity perhaps retained. AG is expected to be derepressed in these organs from an early stage and drives carpel specification, along with CRC.

relevant references:

CRABS CLAW and SPATULA, two Arabidopsis genes that control carpel development in parallel with AGAMOUS - John Alvarez, David R. Smyth

Drews, G. N., Bowman, J. L. and Meyerowitz, E. M. (1991). Negative regulation of the Arabidopsis homeotic gene AGAMOUS by the APETALA2 product. Cell 65, 991–1002

And quoting from “Control of Arabidopsis Flower and Seed Development by the Homeotic Gene APETALA2” K. Diane Jofuku, et al.

In weak, partial loss-of-function ap2 mutants, sepals are homeotically transformed into leaves, and petals are transformed into pollen-producing stamoid organs (Bowman et al., 1989, 1991b). By contrast, in strong ap2 mutants, sepals are transformed into ovule-bearing carpels, petal development is suppressed, the number of stamens is reduced, and carpel fusion is often defective (Komaki et al., 1988; Kunst et al., 1989; Bowman et al., 1991b)

Thus, I think that the figure 5 title “Introducing marginal meristem activity in the sepal largely eliminates the longitudinal developmental gradient” is misleading. The authors are not simply “introducing marginal meristem activity” they are creating a largely carpelloid organ that appears to follow the developmental trajectory that carpels typically follow.

Response #5:

As mentioned above altering the medial/marginal domain without altering carpel identity is impossible with the current experimental tools. We rephrased the manuscript to focus on the tissue specific gradients. For example, we rephrased the main conclusion of the ap2-7 section from “Introducing marginal meristem activity in the sepal largely eliminates the longitudinal developmental gradient” to “Introducing carpel identity into sepal reorient organ differentiation gradients” which in our opinion interprets the data in a less controversial way.

Figure 4 is also challenging to interpret. NPA treatment is also variable and depends on the strength of the treatment. The organs shown in figure 4 display very strong NPA treatment phenotypes. These organs appear to be entirely radialized and entirely lacking valve domains. Previous authors have suggested that these radialized organs represent an organ without ovary domains. The identity of the remaining organ has been hypothesized to be style apically and stipe/or gynophore basally. Patterns of internal vascularization support this. (Auxin and ETTIN in Arabidopsis gynoecium morphogenesis. - Jennifer L. Nemhauser, Lewis J. Feldman, Patricia C. Zambryski or “A model for an early role of auxin in Arabidopsis gynoecium morphogenesis” - Charles Hawkins and Zhongchi Liu, 2014 and “Polar auxin transport together with AINTEGUMENTA and REVOLUTA coordinate early Arabidopsis gynoecium development” Nole-Wilson et al. 2010).

Although the authors document the spreading of the auxin gradient into more basal locations upon NPA treatment, the role of the mediolateral gradient in this process is unclear in this experiment as well. Given the previous observations suggesting an expansion of the length of the style domain upon NPA treatment, one might expect that auxin responses would expand basally due to the transformation of tissue identity. Thus, I find the statement in Line 232 to be overly speculative. (i.e. “These results support the idea that an early differentiation in the valves along the mediolateral axis (likely controlled by CMM signal) is a key event to limit the tip-derived signal from setting up an organ-wide basipetal gradient of cell differentiation.”)

Response #6:

We agree that we might have overinterpreted the NPA data. We now highlight the possibility that the expansion of the longitudinal gradient of growth toward more proximal regions of the valveless gynoecium may be, at least in part, due to the expansion of the style identity. However, the organ wide expansion of PIN3 expression and largely basal direction of cell differentiation in the valveless gynoecium, may still (to some extent) suggest that when early mediolateral gradient of cell differentiation is abolished the differentiation progresses basipetally (now mentioned as perspective in the discussion).

Finally, the elimination of mediolateral gradients in the valveless gynoecium after NPA-treatment supports the idea that cell differentiation gradients are tissue-identity dependent.

I applaud the authors for adding the *crc* spt experiment, but I don't think that it adds much as *crc* is thought to control AG-independent aspects of carpel identity and so you still have a confounding of carpel identity with medial domain.

Response #7:

We now interpret this experiment in light of the change of tissue identity.

Line 270 is overly speculative: (i.e. “These findings confirm that the CMM-dependent, early establishment of mediolateral gradient prevents the progression of late basipetal cell differentiation throughout the entire organ.”)

Similarly, the abstract is overly speculative, in my opinion.

Response #8:

We apologize for these overinterpretations. We now remove these speculations mainly focusing on observed tissue specific developmental gradients.

I am being rather strict and harsh in my assessment. That is in part driven by the high caliber of the Nature Communications standards. It is also driven by my concerns that the data is over interpreted. I think the authors will have success getting this work published, perhaps in a different journal. I think that the paper should be rewritten more-carefully. This will perhaps mitigate the desired impact of the paper, but I don't feel that it is appropriate to make the bold conclusions that are being made here.

One unanswered question that arises from the model:

If there is an interaction between the apical/basal gradient and differentiation in the valve, why is the rate of differentiation between the top of the valve and the bottom of the valve similar. Shouldn't the cells at the bottom of the valve domain act differently from those at the top of the valve. Unless you propose a threshold response effect.

Response #9:

We now remove all simulations from the manuscript.

Minor point –

I don't believe that the data in Extended Data figure 4 is representative of what is happening. If you were to focus up and down through the gynoecium to count the initiating ovule primordia, you would see that almost all the ovule primordia arise nearly simultaneously. There appears to be a miscounting in your approach using just the digital longitudinal section. I believe several groups have reported that the ovule initiation is nearly simultaneous in Arabidopsis. I don't have reference here, but I believe it is work of Chuck Gasser and perhaps Robinson-Beers or Kai Schneitz. You can count them more accurately with chloral hydrate cleared gynoecia.

Response #10: We have already counted ovules, as suggested by the reviewer, utilizing live-image samples where ovules are clearly visible. It has been previously reported that ovules initiate in stage 8 and/or stage 9 of flower development - a period of more than 3 days (Robinson-Beers et al., 1992; Schneitz et al., 1995). Recently, Yu et al., (2020) have clearly shown that ovule initiation is not synchronous: while gynoecium length doubles during ovule initiation, the space between two ovule primordia increases, allowing for new primordia to appear over the period of 3 days. Such asynchronous ovule initiation has also been reported by (Roeder & Yanofsky, 2006).

Reviewer #2 (Remarks to the Author):

The authors have done a good job to soften their interpretations of the data and to reign in claims that were too broad and were poorly supported in the past versions. They have largely removed the problematic modeling aspects.

My remaining concern is how novel and impactful the remaining manuscript is within the field. Given the high bar for publication in Nature Communications, I believe that this study is mid-tier with regard to the additional contribution to the field and a broad relevance to the broader community.